# A first constraint on basal melt-water production of the Greenland ice sheet

Nanna B. Karlsson [1] [✉], Anne M. Solgaard [1], Kenneth D. Mankoff[1], Fabien Gillet-Chaulet[2], Joseph A. MacGregor [3], Jason E. Box [1], Michele Citterio[1], William T. Colgan [1], Signe H. Larsen [1], Kristian K. Kjeldsen [1], Niels J. Korsgaard [1], Douglas I. Benn[4], Ian J. Hewitt [5] & Robert S. Fausto [1]

The Greenland ice sheet has been one of the largest sources of sea-level rise since the early 2000s. However, basal melt has not been included explicitly in assessments of ice-sheet mass loss so far. Here, we present the first estimate of the total and regional basal melt produced by the ice sheet and the recent change in basal melt through time. We find that the ice sheet's present basal melt production is 21.4 +4.4/−4.0 Gt per year, and that melt generated by basal friction is responsible for about half of this volume. We estimate that basal melting has increased by 2.9 ± 5.2 Gt during the first decade of the 2000s. As the Arctic warms, we anticipate that basal melt will continue to increase due to faster ice flow and more surface melting thus compounding current mass loss trends, enhancing solid ice discharge, and modifying fjord circulation.

[1] Geological Survey of Denmark and Greenland, Copenhagen, Denmark. [2] University of Grenoble Alpes, CNRS, IGE, Grenoble, France. [3] Cryospheric Sciences Laboratory, NASA Goddard Space Flight Center, Greenbelt, MD, USA. [4] School of Geography & Sustainable Development, University of St. Andrews, St. Andrews, UK. [5] Oxford Centre for Industrial and Applied Mathematics, University of Oxford, Oxford, UK. ✉email: nbk@geus.dk

Mass loss from the Greenland ice sheet is determined via one of three methods: through estimates of ice volume change from satellite altimetry[1,2], by directly measuring mass changes using gravimetry[3] or by differencing between solid ice discharge and surface mass balance[4,5] (the "input–output" method, the term solid ice discharge refers to the ice mass that exits through flux gates at the margin). The average mass balance of the ice sheet between 2005 and 2015 is $-254 \pm 18$ Gt per year with a spread between different mass balance estimates of 36 Gt per year[6]. Gravity methods implicitly include basal mass loss, while altimetry methods attribute all mass loss to either ice discharge or surface mass loss. Either method provides limited insights into the physical processes leading to the observed change in mass. In contrast, the input-output method relies on accurate process representation of the climatic and dynamic mass-loss terms and thus provides the possibility of predicting future changes. To date, the input-output method has overlooked basal mass balance entirely. Constraining basal melt is important for three reasons. Firstly, uncertainty in the partition of ice-sheet mass loss between surface mass balance and ice discharge, including the failure to acknowledge the basal mass balance term, limits our understanding of changes in ice-sheet mass budget in response to recent climate change. This impedes our ability to capture complex interactions and feedbacks between ice sheets and the climate system. Secondly, the presence or absence of basal meltwater is important for the evolution of the subglacial system[7,8], and recent studies have highlighted the importance of subglacial discharge for modifying the mass loss from marine-terminating glaciers[9,10], it therefore plays an important role for Greenland outlet glaciers' contribution to future sea-level rise[11,12]. Finally, discharge of subglacial water modifies circulation in the fjord systems and may impact nutrient mixing[13,14].

Here, we provide the first estimate of ice-sheet-scale basal melt and its change through the first decade of the 2000s. We consider three sources of basal heat that generate melt (Fig. 1a–c). The first source, the geothermal flux, is assumed to be constant in time, while the other terms, frictional heat and heat from surface melt input to the bed, vary in response to changes in ice dynamics and surface melt, respectively. We quantify basal melt using estimates of geothermal flux, satellite-derived ice-surface velocities, surface and bed topographies, and outputs from an ice-sheet model and regional climate models. We use a multi-year surface velocity composite spanning 1995–2015[15], winter velocity maps from 2000/2001 to 2018/2019[16,17], and average decadal/multi-decadal surface melt-water volumes from 1991–2012[18]. This allows us to construct a baseline basal-melt value against which we can compare likely changes in basal melt rates in the recent past. We assume that all basal melt water is discharged to the ocean or land-margin since the geometry and high surface slopes of the ice sheet preclude the existence of long-term meltwater storage in subglacial lakes[19]. Although studies have found evidence of subglacial lakes[20,21] and "units of disturbed radio-stratigraphy"[22,23], associated volumes are negligible in the context considered here. Similarly, model results indicate that basal freeze-on rates are unlikely to be of significance for the basal mass budget[24]. Our results demonstrate that basal melt is a non-negligible component of the mass balance of the Greenland ice sheet, and that basal melt-water production is likely increasing and will continue to do so in the foreseeable future.

## Results

**Geothermal flux contribution to basal melt**. The heat from the geothermal flux is based on an average of three geothermal flux maps[25–27] and is masked with an independent estimate of where basal ice is likely at pressure melting point[28] (Fig. 1a, black and grey contours). Our estimate of total geothermal basal melt is 5.3 $+ 2.8/-2.2$ Gt per year (Table 1, note that our uncertainty range is asymmetrical and we use '/' to denote upper/lower range). The uncertainty is due to the embedded uncertainties in the geothermal flux estimates as well as the unknown basal temperature of the ice. We find that the difference in ice-sheet-wide basal melt between the geothermal datasets is <10%, however, by including the likely range of geothermal flux based on each dataset's stated uncertainty, the final uncertainty range increases (see methods). Studies suggest that the geothermal flux is generally underestimated in the northeastern (NE) sector due to the presence of a localised "hot spot" under the North East Greenland ice stream[29,30]. Therefore, our estimate comes with the caveat that the contribution from the NE sector is likely larger than the estimate presented here.

Spatially, the basal melt caused by geothermal flux is unevenly distributed (Fig. 1d). The highest melt rates are found in the central eastern (CE) sector where basal melt in a few places exceeds 0.01 m per year. In the CE, SW (southwestern) and SE (southeastern) sectors, melt rates are typically 6–7 mm per year, while melt rates for the remaining sectors are 5 mm per year or less. There is no contribution to the geothermal basal melt in the interior of the ice sheet, where basal ice temperatures are likely below the pressure melting point[28].

**Frictional heat contribution to basal melt**. Frictional heat is produced by ice sliding over the bed. We retrieve an estimate of frictional heat using the Elmer/Ice model, where the complete stress balance is solved ("Full Stokes")[31], and where basal sliding and shear stress are related by a linear friction law[32]. Internal ice temperatures are obtained from a paleo spin-up run[33]. The model uses an anisotropic mesh where the horizontal resolution ranges from ~500 m to ~50 km, but here the original model results have been re-gridded on a 1 km equidistant grid. See also methods for more information Elmer/Ice. Using the present day topography, the spatially-varying friction coefficient is tuned to reproduce the observed surface velocities (Fig. 1b). Thus, the model returns an estimate of basal frictional heating, constrained by surface observations. From this heat estimate we get the resulting basal melt (see methods) and we apply the same mask of basal conditions as used in the geothermal flux calculation[28]. Note that Elmer/Ice predicts basal melt under most of the ice sheet although the basal melt rates are orders of magnitude smaller in masked areas compared to melt rates predicted along the margins. We find that the total basal melt due to frictional heat is $10.9 \pm 2.9$ Gt per year (see methods for a discussion of uncertainties).

Melt from frictional heating is concentrated in areas with high ice-flow velocities i.e. at major glacier outlets (Fig. 1b, e). Most of the basal melt water is drained through large ice streams and several of the major outlets have melt rates orders of magnitude above the melt rates produced by geothermal fluxes. In the slow-flowing interior, friction melt rates are typically at least an order of magnitude lower. In the northern (NO) sector, the outlet of Petermann Glacier is visible as an extended area where friction melt exceeds 0.01 m per year. Near the margin, melt rates can exceed 0.3 m per year. In the NE sector, most of the friction melt is generated by Nioghalvfjerdsfjorden glacier and Zachariae Isstrøm, with rates exceeding 0.2 m per year close to the margin. High friction melt rates are also found in the CE and SE sectors where Kangerlussuaq Glacier and Helheim Glacier cause friction melt in excess of 0.3 m per year. In these three sectors, friction melt rates exceeding 0.01 m per year extend inland. Basal friction as a source of melt is less important in the slow-flowing sectors. In the predominantly land-terminating southwestern (SW) sector, where average velocity is 45 m per year compared to the

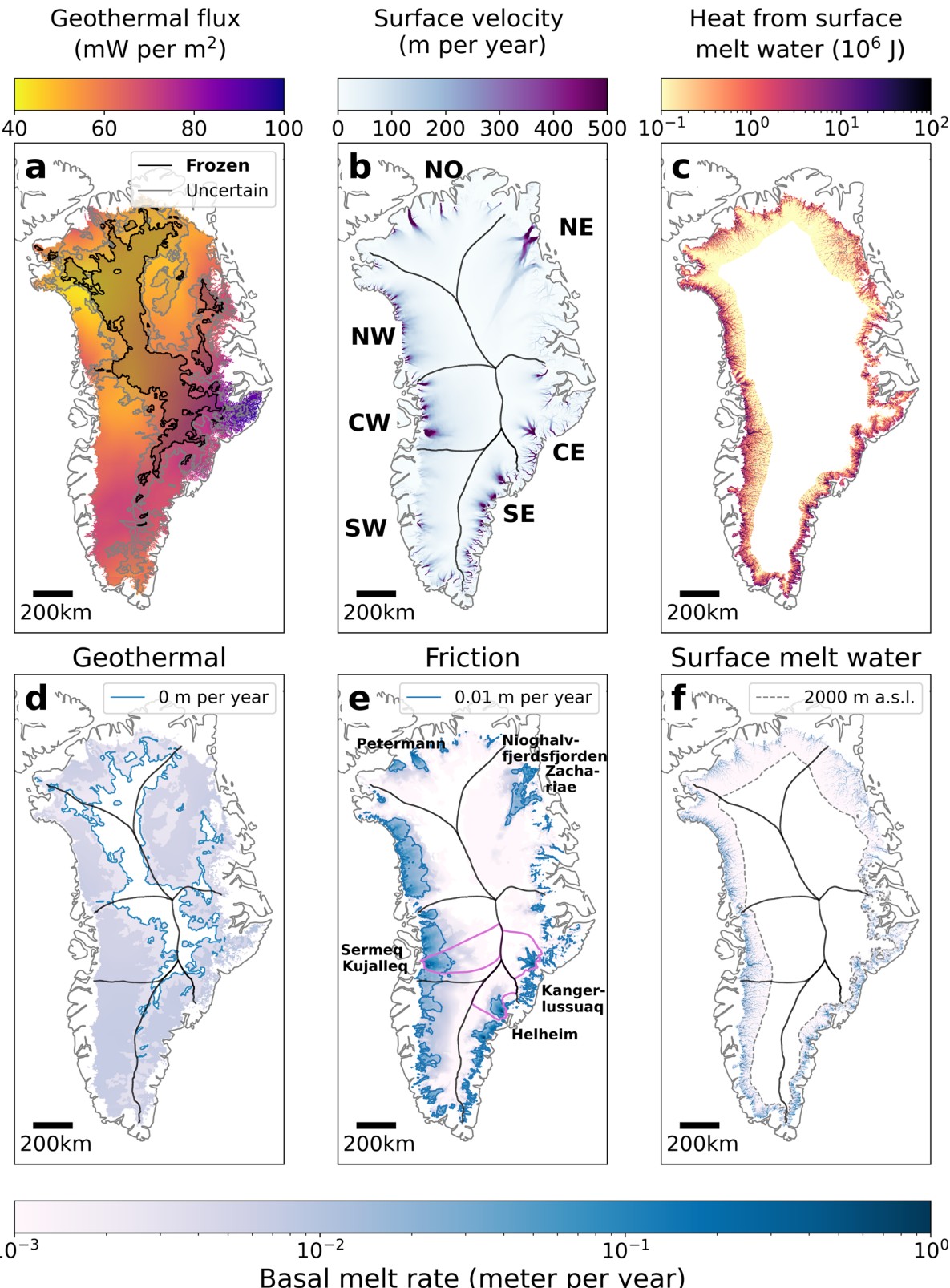

**Fig. 1 Heat sources and resulting basal melt rates. a** Mean geothermal flux from[25–27]. The shaded areas outline where bed conditions are likely frozen (black) or uncertain (grey) based on radar observations and numerical ice-flow models[28]. **b** Surface velocities from multi-year MEaSURES dataset[15]. **c** Heat generated by surface melt-water infiltration. **d** Basal melting from geothermal heating. Blue contours outline the 0 m per year extent. **e** Basal melting from frictional heating. Purple outlines show the glacial catchments of Sermeq Kujalleq, Kangerlussuaq and Helheim Glacier[55]. Blue contours outline the $10^{-2}$ m per year extent. **f** Basal melting from surface water heating. Dashed grey contours outline the 2000 m above sea level elevation. **d–f** have the same logarithmic scalebar.

**Table 1 Basal melt from the three heat terms and the total basal melt.**

| Sector | Geothermal (Gt per year) | Friction (Gt per year) | Surface water (Gt per year) | Total melt (Gt per year) |
|---|---|---|---|---|
| Central east (CE) | 0.5 + 0.5/−0.3 | 1.2 ± 0.3 | 0.5 ± 0.2 | 2.3 + 0.6/−0.5 |
| Central west (CW) | 0.7 + 0.3/−0.2 | 2.4 ± 0.6 | 0.7 ± 0.2 | 3.9 + 0.7/−0.7 |
| Northeast (NE) | 1.3 + 0.6/−0.5 | 1.0 ± 0.3 | 0.5 ± 0.1 | 2.8 + 0.7/−0.6 |
| North (NO) | 0.4 + 0.3/−0.3 | 0.6 ± 0.2 | 0.4 ± 0.1 | 1.5 + 0.4/−0.3 |
| Northwest (NW) | 0.6 + 0.2/−0.2 | 2.1 ± 0.6 | 0.8 ± 0.3 | 3.5 + 0.7/−0.6 |
| Southeast (SE) | 0.7 + 0.5/−0.3 | 2.2 ± 0.6 | 0.8 ± 0.3 | 3.7 + 0.8/−0.7 |
| Southwest (SW) | 1.2 + 0.4/−0.4 | 1.3 ± 0.4 | 1.4 ± 0.4 | 3.9 + 0.7/−0.7 |
| Total | 5.3 + 2.8/−2.2 | 10.9 ± 3.0 | 5.2 ± 1.6 | 21.4 + 4.4/−4.0 |

The friction heat term is based on ice-velocity data spanning 1995–2015 while the surface melt-water heat term spans 1995–2010.

61 m per year Greenland-wide average, friction melt does not exceed 0.2 m per year except in a few locations near the ice margin. The central western (CW) sector has the largest areal extent of high friction melt rates and undergoes melt rates above 0.4 m close to the margin in several places. High friction melt in the CW sector is in part due to Sermeq Kujalleq (Jakobshavn Isbræ), one of Greenland's largest and fastest outlet glaciers. In contrast, the northwestern (NW) sector contain numerous smaller glaciers but combined they also create a large area where melt rates exceed 0.01 m per year.

**Surface melt water heat contribution to basal melt.** Finally, we consider heat generated by surface melt water as it infiltrates the subglacial system (Fig. 1c). We convert the gravitational potential energy of surface melt water into heat, which melts open sub-glacial conduits as water flows through the ice sheet, assuming that all water reaches the bed. In contrast to the geothermal and frictional terms, melt due to surface melt water is focussed in conduits and thus highly localised. This entails that water is allowed refreeze locally due to supercooling as described in[34]. We further assume that the water only penetrates to the bed at altitudes below 2000 m above sea level. This heat source has been calculated in previous studies[34] using surface water volumes from a regional climate model[35] but not translated directly into basal melt rates. Here, we use a recently published surface melt-water estimate based on an average of 13 regional climate models[18]. We estimate that the average basal melt due to surface melt-water injection was 5.2 ± 1.6 Gt per year in 1990–2010. Uncertainties stem from the reported 30% variability between regional climate model results. Note that there is significant variation between models on a sector-by-sector basis.

The basal melt due to surface melt water is focussed in areas where surface melt occurs, and where the water is subjected to large hydropotential gradients as it flows along the ice-sheet bed (Fig. 1f). The heat from the surface melt causes substantially higher basal melt rates than the geothermal flux along the high-gradient ice-sheet periphery. The basal melt rates due to surface melt water exceed 0.05 m per year in a few places along the margin but the bulk of the sectors have melt rates below 0.5 mm per year. In contrast to the geothermal and frictional terms, the melt due to surface melt water is likely to be focussed in the conduits and thus highly localised. The values reported above represent an average over 1 km grid cells masking the fact that melt rates may vary orders of magnitude over sub-kilometre distances.

**Total basal melt on regional and local scales.** Our baseline basal melt discharge is estimated at 21.4 + 4.4/−4.0 Gt per year, equivalent to 4.5% of the annual solid ice discharge (average of 1986–2018 ice discharge[5]). The basal melt also corresponds to more than half of the annual discharge of Sermeq Kujalleq

(average of 1986–2018), the largest single Greenlandic glacier contributing to sea-level rise[5]. At ice-sheet scale, basal melt is primarily caused by frictional heating (51%), with surface-melt water heat and geothermal heat as secondary contributors (24% and 25%, respectively, Fig. 2a and Table 1). The individual contributions from each of the heat terms vary for the different ice-sheet sectors depending on local geothermal flux anomalies and surface melt-water volumes. For example, in the slow-flowing SW sector the relative contributions from the three heat terms approach parity, while friction heat dominates in the CW sector (Table 1).

The largest basal mass loss occurs in the CW and SW sector (3.9 ± 0.7 Gt per year), followed by the SE sector (3.7 + 0.8/−0.7 Gt per year) and the NW sector (3.5 + 0.7/−0.6 Gt per year). The NO sector has the smallest basal mass loss (1.5 + 0.4/−0.3 Gt per year) due to a combination of low friction melting and small volumes of surface melt water. The largest mass loss due to surface melt-water heat occurs in the SW sector, while the largest losses due to friction heat and geothermal flux occur in the CW and NE sectors, respectively (Table 1). We note that in order to represent basal mass loss on a sector basis, the subglacial drainage basins are assumed identical to the glaciological drainage basins. On drainage-basin scales, we only present the basal melt discharge for three of the largest glaciers (by discharge and flux gate size): Sermeq Kujalleq, which discharges into Qeqertarsuup tunua (Disko Bay), Kangerlussuaq Glacier that discharges into Kangerlussuaq Fjord and Helheim Glacier that terminates in Sermilik Fjord. Here, we calculate the individual subglacial basins using the hydropotential assuming that the subglacial water pressure is at ice overburden pressure[36]. We estimate that at present, the basal melt water flux from Sermeq Kujalleq is 1.6 ± 0.5 Gt per year and 41% of the basal melt water from the CW sector exits through Sermeq Kujalleq into Qeqertarsuup tunua. At Kangerlussuaq Glacier the basal melt discharge is 0.8 ± 0.2 Gt per year, corresponding to 35% of the basal melt water in the CE sector. Finally, we find that for Helheim Glacier, the basal melt discharge is 0.9 ± 0.3 Gt per year (24% of discharge in SE sector).

**Temporal evolution of frictional and surface melt-water heat.** Above, we reported on a baseline value that represents a multi-decadal average. However, as ice dynamics and surface mass balance respond to changes in climate, by extension the basal-melt contributions from friction heat and surface melt-water heat must also change.

The ice sheet underwent a general speed-up during the 2000s[4,5] and here we investigate its potential effect on the friction melt. In order to obtain annual friction-melt estimates, we need to use a simplified description of the ice dynamics. This is necessary because while Elmer/Ice returns high-resolution insights into the basal melt rates, it comes with substantial computational expense. Instead, we use a simplified approach where the basal sliding is

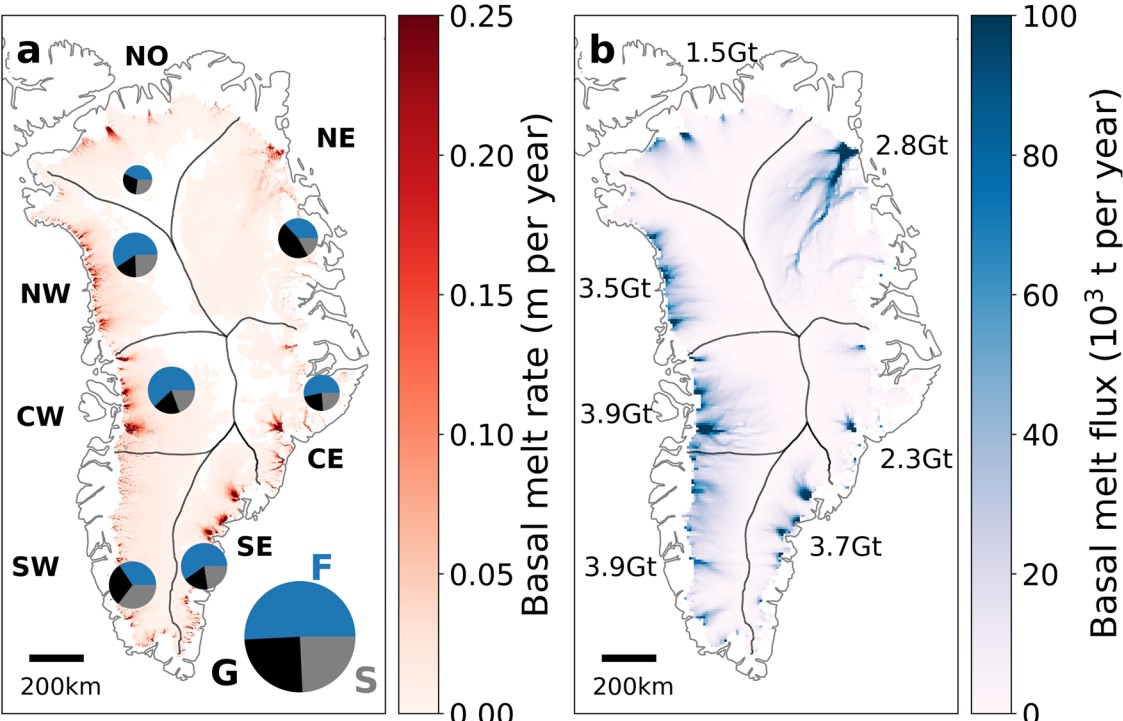

**Fig. 2 Total basal melt rates and the resulting flux. a** Basal melt rates. Pie charts show the contribution from the different heat terms: friction heat (F, blue), geothermal flux (G, black) and viscous heat dissipation from surface melt water (S, grey). Size of circles indicate the total basal melt discharge from each sector. **b** Flux of basal melt water. Numbers show the total basal melt discharge for each sector.

assumed equal to the difference between observed winter surface velocities and deformational (creep) velocities[37] (see methods). The use of winter velocities entails that we are underestimating the friction heat while the simplified approach introduces additional uncertainties (see methods). We find that the basal melt from our simplified approach is 31% higher compared to the basal melt from the Full Stokes approach. The simplified stress-balance overestimates the basal melt in all sectors (except the CE sector) but the difference is not evenly distributed between sectors with the largest differences in the NE region (59%) and NW sector (52%) (see methods and Supplementary Note 1). The reason for the large discrepancies is likely the inability of the simplified approach to capture the complex flow regime of the Northeast Greenland ice stream in the NE sector, and the topography of numerous small outlet glaciers in the NW sector. Our findings are consistent with a recent study showing that the simple approach overestimates the basal stresses compared to the Full-Stokes solution[32]. In addition to the uncertainty imposed by the simplified stress-balance, other uncertainties include unknown temperatures of the basal shear layer and the uncertainties from velocity datasets (see methods for a detailed discussion of the uncertainties). In particular, we assume that the basal shear stress remains constant despite the velocities changing (a reasonable approximation on the coarser scale of this approach, since overall force balance must be maintained, but which would not necessarily be true on a local scale). Using this simplified approach, we estimate that the friction melt has increased from 10.6 ± 4.3 Gt in winter 2000/2001 to 11.8 ± 4.5 Gt in winter 2017/2018, corresponding to an increase of 10% (Fig. 3). The uncertainty range is mainly due to parameters that are constant in time thus we posit that the reported increase is a consequence of increased ice-flow velocities. A linear regression through the velocity datasets from 2005/2006 through 2017/2018 indicates that basal friction discharge has increased by 0.09 + 0.04/−0.03 Gt per year. Note that basal shear stress is assumed to

remain constant. Over most of the ice sheet, glacier geometry (and hence driving stresses) did not change significantly during our study period, implying near-constant resisting stresses on the large spatial scale used in our simplified model.

The surface melt-water volume exhibits high interannual variability and thus constructing a regression line is less meaningful. Instead, we consider the decadal averages 1991–2000 and 2001–2010. We find that basal melt due to surface melt water increased from an average of 3.5 ± 1.1 Gt per year in 1991–2000, to an average of 6.0 ± 1.8 Gt per year in 2001–2010 (Table 2). This corresponds to a 70% increase in basal melt directly caused by increased volumes of surface melt water. The basal melt for all sectors increased by more than 50% with the largest increase in the NW sector of 110%. In order to estimate future change in basal melt due to increased surface melt water, we consider surface melt for 2012. While this was an extreme melt year in the context of present-day melt rates, it is likely that such melt-water volumes will become more common in the future[35]. Using 2012 surface melt water volumes as an analogue of the likely increased future melt, we get basal melt rates of 10.0 ± 3.0 Gt per year, corresponding to an increase of 4.8 Gt or more than 90% compared to our baseline value for 1995–2010. The largest increase is found in the NE sector (149%) but all sectors experience an increase in basal melt caused by surface melt water (Table 2). In the NE, NO and SW sectors, the basal melt rates from 2012 surface melt water exceed the baseline friction-melt term implying a shift in principal basal melting process. Overall, in the future, basal melt due to heat from surface melt water is likely to become as important as friction melt for ice sheet mass loss.

Assuming that the friction-melt term from winter 2000/2001 is representative of the preceding decade, we estimate that the total basal melt production has increased from 19.4 + 6.0/−4.7 Gt per year in the 1990s to 23.1 + 6.1/−4.9 Gt per year in the following decade. The change is due to an increase in friction-induced basal

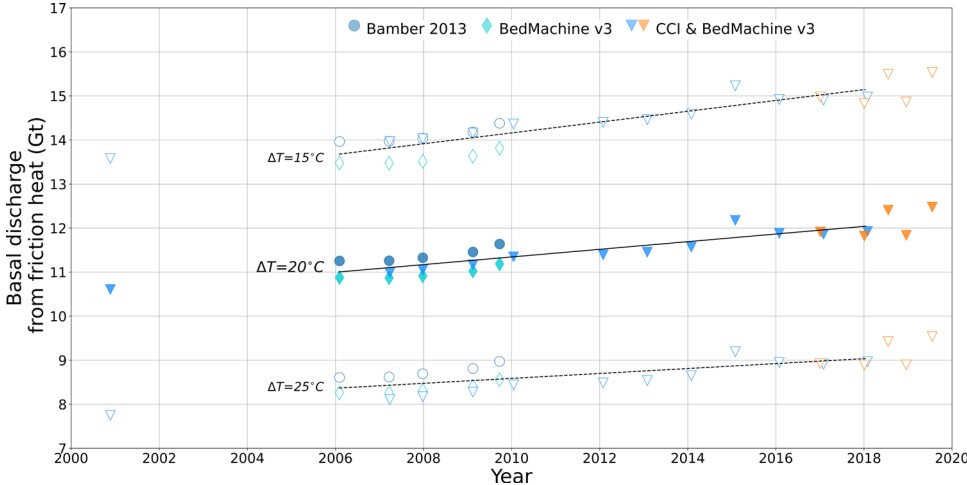

**Fig. 3 Basal melt discharge due to friction heat from winter 2000/2001 through to summer 2019.** Blue and turquoise colours indicate results based on the gap-filled MEaSUREs dataset (see methods). Orange colours indicate that results are from the PROMICE Sentinel-1 derived velocities. Black line is best linear fit through the MEaSUREs datasets (from the years 2005/2006, 2007/2008, 2008/2009, 2009/2010, 2012/2013, 2014/2015, 2015/2016 and 2016/2017), dashed black lines represent best linear fit if internal ice deformation temperatures are offset by ±5 °C. The shape of the points indicate origin of surface and bed topographies.

**Table 2 Basal melting in Gt per year due to surface melt-water heat for decadal averages 1991–2000 and 2001–2010, and for 2012.**

| Sector | Surface water 1991–2000 (Gt per year) | Surface water 2001–2010 (Gt per year) | Surface water 2012 (Gt per year) |
|---|---|---|---|
| Central east (CE) | 0.4 ± 0.1 | 0.6 ± 0.2 | 0.9 ± 0.3 |
| Central west (CW) | 0.5 ± 0.2 | 0.8 ± 0.3 | 1.4 ± 0.4 |
| Northeast (NE) | 0.3 ± 0.09 | 0.6 ± 0.2 | 1.2 ± 0.3 |
| North (NO) | 0.3 ± 0.08 | 0.5 ± 0.1 | 0.9 ± 0.3 |
| Northwest (NW) | 0.5 ± 0.1 | 1.0 ± 0.3 | 1.6 ± 0.5 |
| Southeast (SE) | 0.6 ± 0.2 | 0.9 ± 0.4 | 1.4 ± 0.4 |
| Southwest (SW) | 1.0 ± 0.3 | 1.5 ± 0.5 | 2.6 ± 0.8 |
| Total | 3.5 ± 1.1 | 6.0 ± 1.8 | 10.0 ± 3.0 |

Note the substantially higher melt in 2012 due to large volumes of melt water.

melt of 0.4 ± 4.8 Gt (from 10.6 ± 4.3 Gt in winter 2000/2001 to 11.0 ± 2.1 Gt (mean of winters 2005/2006–2009/2010 using BedMachine topography)), and in basal melt due to surface melt water of 2.5 ± 2.1 Gt. This corresponds to a total increase of 2.9 ± 5.2 Gt.

## Discussion

We have shown that the volume of basal melt water from the Greenland ice sheet is a non-negligible part of the total mass budget. With a total mass balance of −254 ± 18 Gt per year[6], basal melt discharge is presently equivalent to 8% of this imbalance but has hitherto not been included in input-output estimates of total mass loss. Basal melt will change as the Greenland ice sheet responds to a warming climate. The frictional heat will increase if the areal extent of the fast-flowing regions expand, leading to an increase in basal melt production. However, the impact of climate change on ice-stream dynamics is complex and thus, we cannot predict by how much the friction term will increase. Based on the recent past (Fig. 3), if glaciers continue to accelerate, basal melt water production may increase by ~0.1 Gt every year into the foreseeable future. Heat transported by surface

melt water will increase with greater melt-water production, which will likely increase melt-water delivery to the bed especially in the ablation zone. Under a high-emissions scenario, this melt source will experience a substantial 5-to-7-fold increase by 2100[34]. Thus, the overall mass loss associated with increased surface melt will be further enhanced by the additional basal melt caused by the viscous heat dissipation from the surface melt water.

Basal melting may also have a large effect on fjord processes and ice-ocean interaction. During winter, the basal melt discharge that stems from frictional heat and geothermal flux is generated independently of surface melt. Thus, the basal melt introduced and quantified here is a primary source of winter subglacial discharge, and this influx of winter basal water is poorly understood and sparsely measured[38]. Biological productivity is affected by subglacial discharge that modifies mixing in the fjords[14,39], but the impact of increasing winter freshwater on Arctic fjord environments is as-yet unknown. Studies suggest that winter basal melt discharge may drive year-round submarine meltwater plumes leading to persistent ice-front melting, and that basal melt discharge may pull in warm water from the Atlantic further enhancing frontal melt rates[40]. Finally, recent and future increases in basal melting likely have a non-linear effect on ice-sheet discharge. The projected contribution to sea-level rise from the Greenland ice sheet is markedly larger when subglacial discharge is increased, and this effect is comparable to the increase caused by rising ocean temperatures[11]. Thus, an increase in basal melt will likely further compound mass loss from marine-terminating glaciers.

## Methods

**Geothermal heat.** We use the average geothermal flux from three published studies[25–27]. Note that one of the datasets (Fox Maule[25]) does not cover the southern tip of Greenland so in this region, the average geothermal flux map is based on only two datasets ([26] and [26]). We calculate the resulting melt rates from the geothermal heat assuming that the ice is at pressure melting point[37].

$$\dot{b}_m = \beta \frac{E_b}{\rho_i L} \qquad (1)$$

where $E_b$ is available energy at the bed, here the geothermal flux, $\rho_i$ is the density of ice, and $L$ is the latent heat of fusion. The $\beta$-parameter indicates the basal conditions. We construct $\beta$ using a map of estimated basal conditions based on a combination of radar observations and model studies[28], where bed conditions were

classified as either "likely frozen", "uncertain" or "likely thawed". Here, we assume that $\beta = 0$ where grid cells are assigned as "frozen", $\beta = 1$ where grid cells are "thawed", and $\beta = 0.5$ for all "uncertain" grid cells.

Two sources contribute to the uncertainty of our estimate: The uncertainty of the geothermal flux maps and the unknown basal temperature. We assess the former by considering the spread in geothermal flux between the maps. Here, we adapt the approach of[41] and define the standard deviation of the geothermal flux $\sigma_G$ as

$$\sigma_G = \sigma[G_1 + \delta_1, G_1 - \delta_1, G_2 + \sigma(G_2), G_2 - \sigma(G_2), G_3 + \delta_3, G_3 - \delta_3] \quad (2)$$

The uncertainty, $\delta$ of the first dataset[42], $G_1$, is stated as ranging from 21 to 27 mW m$^{-2}$ [25], where we choose the higher value. The second dataset[26], $G_2$, does not supply an uncertainty and lacking any other information we use the standard deviation that is given for each data point. The third dataset[27], $G_3$, supplies an uncertainty. We use the standard deviation to calculate the basal melt from the spread $\bar{G} + \sigma_G$ and $\bar{G} - \sigma_G$, in addition to the basal melt from the mean geothermal map $\bar{G}$. This returns an uncertainty of ±21% in total basal melt. On a catchment-scale basis, this change varies with the largest spread in the SE sector of 34%, while the largest spread in absolute values is 0.29 Gt per year from the SW sector (see Supplementary Note 2).

The second uncertainty is the unknown basal temperature of the ice. We continue to make use of the results from[28] and construct two scenarios: a thawed scenario where we assume that all regions classified as uncertain are thawed (i.e. we change all areas where $\beta = 0.5$ to $\beta = 1$), and a frozen scenario where we assume that all uncertain regions are frozen (i.e. we change all areas where $\beta = 0.5$ to $\beta = 0$). We obtain the final uncertainties by considering two end members: 1) a "warm" scenario where all uncertain areas are assumed to be thawed and where the geothermal flux equals $\bar{G} + \sigma_G$, and 2) a "cold" scenario where the base is frozen in uncertain areas and where the geothermal flux is $\bar{G} - \sigma_G$. This gives an upper value of basal melt of 8.1 Gt per year and a lower value of basal melt of 3.1 Gt per year. Thus, the basal melt due to geothermal flux is 5.3 + 2.8/−2.2 Gt per year (see Supplementary Note 2 for all ranges for each sector and maps showing the resulting basal melt for the different scenarios considered here).

**Frictional heat: Elmer/Ice model.** The first estimate of frictional heat is obtained with the Elmer/Ice model, which is a Full Stokes ice-flow model resolving all stresses[31,32]. The ice-flow model uses the GIMP digital elevation model (Greenland Ice sheet Mapping Project[43]), and ice thicknesses and bed topography from Bed-Machine v3 calculated using a mass-conservation method[44]. We apply an ice cover mask[45] in order to remove local ice caps and glaciers. Elmer/Ice uses an anisotropic mesh optimised to capture velocity and thickness variations, and insure a high resolution in the first 40 km from the ice-sheet edges. The resulting horizontal resolution ranges from ~500 m to ~50 km. Original model results have been re-gridded on a 1 km equidistant grid for the post-processing (http://elmerfem.org/elmerice/wiki/doku.php?id=eis:greenland). The internal ice temperature field comes from a paleo spin-up of the SICOPOLIS model[33]. The Elmer/Ice model is inverted in order to minimise the misfit between modelled and observed surface velocities. The inverse method uses a multi-year average of the surface velocity in 250 m resolution from the MEaSUREs (Making Earth System Data Records for Use in Research Environments) Greenland Ice Velocity data based on data from RADARSAT-1, ALOS, TerraSAR-X/TanDEM-X and Sentinel-1A and -1B[15,16,46].

The model is computationally expensive which makes it unfeasible to run an ensemble of models to obtain formal estimates of the uncertainties. Instead, we investigate the uncertainties associated with our simplified stress-balance model and based on insights from these experiments, we estimate the uncertainty of the Elmer/Ice output.

**Frictional heat: shallow-ice approximation.** The second model that we use to obtain the frictional heat term is a simplified stress-balance equation, the shallow-ice approximation[37], coupled with the velocity observations to calculate the basal sliding velocity. In this model, we use the surface topography from the Climate Change Initiative (CCI, http://cci.esa.int/) derived from the ArcticDEM (Arctic Digital Elevation Model[47]) based mainly on the WorldView 1-3 satellites. This gives a long temporal baseline from 2007 until present day. We combine the CCI surface elevation with the BedMachine v3 bed topography data[44]. We apply an ice cover mask[45] in order to remove local ice caps and glaciers. Ice-flow velocities are obtained from two sources: MEaSUREs and the PROMICE (Programme for Monitoring of the Greenland ice sheet) velocity product based on Sentinel-1A and -1B[17,48]. The MEaSUREs velocity maps cover the periods from winter 2000/2001 to winter 2017/2018 although the coverage is not continuous: Velocity maps are not available from 2001/2002 to 2004/2005. Only the latest velocity maps are complete so in order to get better coverage for our estimate of temporal changes, we apply the same methodology as described in[5] and linearly interpolate missing values in time. We do not interpolate spatially since spatial changes are most likely larger than temporal changes for any given point. Data at the beginning or end of the time series are back- or forward-filled with the temporally nearest value for that grid cell.

The PROMICE dataset spans winter 2016/17 to winter 2018/19 and is based on intensity offset tracking. Here, the data coverage is near complete and no interpolation is necessary. We note that the PROMICE maps overestimate the velocities in the interior of the ice sheet where MEaSUREs relies on the more accurate InSAR.

The shallow-ice approximation employed here is based on the assumption that on spatial scales over several ice thicknesses, ice flow can be assumed to consist of two components: deformational velocity $u_d$ (at times also referred to as creep velocity) and basal sliding $u_b$[37]. Thus the total velocity is

$$u = u_d + u_b \quad (3)$$

and here we assume that $u$ is equivalent to the observed surface velocity $u_o$. Our method thus retrieves the basal velocity using the observed surface velocity and the calculated deformational velocity. Theoretically, the surface velocity due to deformation is[37]

$$u_{s,def} = \frac{2A(T)}{n+1}\tau_b^n H, \quad (4)$$

where $A(T)$ is the flow law parameter, $H$ is ice thickness, $n$ the flow law exponent, and $\tau_b = \tau_d = \rho g H \nabla s$, where $\rho_i$ is ice density, $g$ is gravity and $\nabla s$ is the surface gradient. We perform this calculation on a 10 km grid where ice surfaces have been smoothed by a 20 km running mean (in order to smooth over several ice thicknesses[37]). From the theoretical deformational velocities we thus get our basal sliding velocity

$$u_b = u_o - u_{s,def} \quad (5)$$

and from this we can directly calculate the frictional heat and thereby the melt rate, assuming that the temperature of the ice is at pressure melting point:

$$\dot{b}_m = \frac{u_b \, \tau_b}{\rho_i L} \quad (6)$$

where $L$ is latent heat of fusion of ice at 0 °C.

**Frictional heat: uncertainties.** In the following, we discuss and quantify the uncertainties relating to our frictional heat estimate. We first present the uncertainties associated with the shallow-ice approximation and use the insights to estimate uncertainties for Elmer/Ice.

A main uncertainty is the unknown ice temperatures. The flow law parameter $A(T)$ depends on temperature (cf. Eq. (4)). Since most of the deformation takes place in the lower 20% of the ice column, the appropriate value for $A$ in our case is probably closer to the temperature at the bed than the average temperature of the ice column. We use internal ice temperatures derived from radar-attenuation values[49] to calculate the deformational velocities, and add a constant offset of 20 °C (see Supplementary Note 4) to capture temperatures in the lower 20% of the ice column where ice is warmer than the overlying ice[37]. In order to investigate the uncertainties due to poorly constrained internal temperatures, we vary our constant temperature offset by ±5 °C. We chose ±5 °C as a likely uncertainty range because comparison between the internal temperature and estimated basal conditions reveals that changing the offset by more than −5 °C returns cold conditions in areas that are likely thawed at the bed[28], while changing the offset by more than +5 °C returns warm conditions in areas that are likely frozen at the bed[28]. We find that a change in temperature of +5 °C leads to a change in basal melt from frictional heat by −25%, conversely a change in temperature of −5 °C leads to a 25% increase in basal melt (for the 2018/2019 velocity dataset).

We rely on observed surface velocities to infer the basal sliding, and thus our results are also affected by uncertainties in the velocity data. To translate the velocity uncertainty into friction-melt uncertainty, we perturb all velocity data points by a randomly selected number between − 1 and 1 multiplied with the standard deviation for the point. In this way, we generate 1000 perturbed velocity maps for each MEaSUREs dataset from the years 2005/2006, 2007/2008, 2008/2009, 2009/2010, 2012/2013, 2014/2015, 2015/2016 and 2016/2017. We then calculate the friction melt for each perturbed velocity map and find that this leads to a distribution of friction melt values where 95% of values deviate less than ±1% from the mean value, and we therefore assign an uncertainty of ±1% caused by uncertainties in the velocity datasets.

We primarily make use of winter velocities potentially leading to an underestimation of annual basal melt rates since summer velocities are typically higher. We use winter velocities due to the lack of complete maps from summer observations. However, with the recent launch of Sentinel-1, it is possible to construct complete summer velocity maps, and we have included two maps from summers 2018 and 2019. The resulting basal melt rates are 5% higher for these summer maps due to the increased ice-flow velocities. We note that in our simplified stress-approximation, an increase in surface velocity translates directly into an increase in friction heat because we assume that the resistance to sliding over the bed is constant regardless of velocity. Assuming that summer velocities are representative for at most 50% of the year, we estimate that exclusively using winter velocities leads to an underestimate of 2.5%.

Due to the simplicity of the shallow-ice approximation, we are also able to explore the impact of using different surface and bed topographies. We use two

different bed topographies and three different surface elevation datasets. We use the kriging-based bed topography published in 2013[50] and the bed topography from BedMachine v3. In addition to the surface topography from the Climate Change Initiative, we use the two GIMP-derived surface topographies from[50] and BedMachine that spans a time period between 20 February 2003 to 11 October 2009. Using the basal melt results from winters 2006/2007, 2007/2008 and 2008/2009, we investigate the impact of the difference in topographic datasets. We find that the difference is less than 4% and typically of the order of 2% depending on temperature offset. We use 4% as a conservative upper bound.

Assuming that the uncertainties discussed above are independent, we use a simple error propagation (square root of the sum of squares) and get an uncertainty of ±27%. We assume that this uncertainty range is applicable to both the Elmer/Ice and the shallow-ice approximation models. While Elmer/Ice makes use of temperatures from a paleo spin-up run, its temperature field is still subject to uncertainties, and we consider that a ±5 °C uncertainty range is not unlikely.

In addition to the uncertainties listed above, studies have shown that deformation predicted by the shallow-ice approximation deviates from observations particularly when sliding is present[51] implying that our predicted basal sliding is incorrect. Furthermore, the shallow-ice approximation limits our horizontal resolution and may not resolve all the narrow (below 20 km wide) and fast flowing outlet glaciers. Comparison with outputs from the Elmer/Ice model shows that the simplified stress-balance leads to an overestimation of basal melt rates of 31%. Note that in this comparison we use the same temperature and surface velocity fields in both models so that the difference is mainly due to differences in resolution and stress approximation. The overestimation is particularly pronounced in areas with high surface velocities (e.g. Sermeq Kujalleq) and complex stress regimes (the Northeast Greenland Ice Stream). See also Supplementary Fig. 1 for a map highlighting the differences. The largest differences are found in the NE region (59%) and NW sector (52%), while the difference for other sectors vary between −4% and 38%. Thus, our simple model leads to an overestimation of basal melt rates relative to the Full Stokes model. We assign a total uncertainty to the values calculated with the shallow-ice approximation of 41%, based on error propagation of the 31% uncertainty discussed here and the 27% uncertainty derived in the sections above. Interestingly, recent observations of a borehole in western Greenland found that ice flow was dominated by sliding in spite of slow ice flow[52]. Our simple analysis infers negligible basal sliding in slow-flowing areas implying that we might be underestimating frictional heat in slow-flowing areas. However, the contribution of basal melt from slow-flowing area is likely orders of magnitudes smaller than the basal melt generated in fast-flowing areas, implying that this underestimation is within our stated uncertainty range.

We use the shallow-ice approximation primarily to estimate the temporal change in basal melt, making use of the simplified ice-flow model in order to be able to conduct more model runs. Although the uncertainty of each individual year is 41%, we postulate that the uncertainty in the change in basal melt is significantly smaller. Below, we outline the reasoning behind this conjecture. Again we note that our simplified stress-approximation assumes that the basal stress is constant.

We assume that the internal ice temperature is constant in time and thus the uncertainty from the unknown internal temperature is negligible when considering the change in basal melt. We also assume that the uncertainties imposed by the simplified stress balance and the low resolution are constant in time. This assumption is based on the consideration that while the general speed up of the ice sheet should lead to faster and potentially more widespread fast flow, the extent of areas exhibiting complex stress regimes is likely to remain the same, and thus the difference between a Full Stokes calculation and a shallow-ice approximation remains constant.

Instead, uncertainties for the change in friction melt are firstly, based on the difference in slope for the three temperature offsets (black lines in Fig. 3) and secondly on the uncertainty from the MEaSUREs velocity datasets. It should be noted that gaps in the velocity fields typically are back-filled with data points from later observations where velocities are likely higher, thus we are underestimating the temporal change in basal melt due to the back-filling. Note, that we only use datasets from years 2005/2006, 2007/2008, 2008/2009, 2009/2010, 2012/2013, 2014/2015, 2015/2016, and 2016/2017 to calculate the regression line shown in Fig. 3 because these datasets have less than 25% of back-filled grid points. The difference in slope for the three temperature offsets can be found straightforwardly by subtracting the slopes of the regression line. The total uncertainty is then found with simple error propagation (square root of the sum of squares for the two terms).

**Subglacial water routing and viscous heat dissipation.** We estimate the surface melt water contribution using previously published methodology[34] where heat estimates are derived from runoff values from the GrSMBMIP project (Greenland Surface Mass Balance Model Intercomparison Project). The GrSMBMIP project compiles results from 13 regional climate models during 1980–2012 CE and we use the average values from all 13 models. The reported spread in modelled surface melt water volumes is 30% and we use this range as our uncertainty.

We assume that the subglacial water pressure is equal to the overburden pressure and that the subglacial water follows the steepest gradient of the

hydropotential[36]Φ

$$\Phi = \rho_w g z_b + \rho_i g(z_s - z_b), \quad (7)$$

where $\rho_w$ is the density of water, $\rho_i$ is the density of ice, and $z_b$ and $z_s$ are the elevations of bed and surface topography, respectively.

As the basal melt water travels through the subglacial system it follows the hydropotential gradient, and energy is released. This energy $Q$ is tracked and depends on the volume of water $V$, the change in hydropotential, and the change in phase transition temperature (last term)

$$Q = V(\nabla \Phi - C_T c_p \rho_i \rho_w g \nabla(z_s - z_b)), \quad (8)$$

where $C_T$ is the Clausius–Clapeyron slope ($8.6*10^{-8}$ K Pa$^{-1}$), and $c_p$ the specific heat of water 4184 J K$^{-1}$ kg$^{-1}$. We route the water using Eq. (7), assuming that water moves to the neighbouring cell with the lowest hydropotential. The routing algorithm is an industry-standard GIS (Geographic Information System) hydrological routing algorithm in GRASS GIS (Geographic Resource Analysis Support System GIS).

We assume that all potential energy is converted to heat[34], that surface water immediately penetrates to the bed and that the englacial water is at the pressure melting point, meaning that the viscous heat dissipation contribution to basal melt is effectively equivalent to the ice volume melted to form the en- and subglacial conduits[53]. Note that we also assume that the water only penetrates to the bed at altitudes below 2000 m above sea level. Tests using equilibrium line altitude instead of the 2000 m elevation contour found that the resulting change in basal melt was less than 5%[34]. The viscous heat dissipation is the sole reason why the surface melt water increases the basal melt rates. We also keep track of the energy budget as meltwater is routed through the hydrological system, producing additional meltwater. This additional meltwater in turn may melt out more water in a positive feedback. We do not resolve the location of individual conduits explicitly and thus lacking information on their exact location, we assume that they are situated at the bed, and we calculate the potential energy of this additional melt. Locally, this leads to less than 1% increase in basal melt rates.

## Data availability
All basal melt maps are available at the GEUS Dataverse, https://doi.org/10.22008/FK2/PLNUEO[54]. Velocity maps constructed through the PROMICE programme using Sentinel-1 are available at the PROMICE website (www.promice.dk).

## Code availability
Code showing examples of how to generate Figs. 1d–f and 2a are available at the GEUS Dataverse website[54].

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

## Acknowledgements

PROMICE is funded by the Geological Survey of Denmark and Greenland (GEUS) and the Danish Ministry of Climate, Energy and Utilities under the Danish Cooperation for Environment in the Arctic (DANCEA), and is conducted in collaboration with DTU Space (Technical University of Denmark) and Asiaq, Greenland. The Elmer/Ice model computations presented in this paper were performed using the GRICAD infrastructure (https://gricad.univ-grenoble-alpes.fr), which is supported by Grenoble research communities. The authors gratefully acknowledge insights from S. Rysgaard (Aarhus University, Denmark) and M. Oksman (GEUS) on marine nutrients and primary production.

## Author contributions

N.B.K. conceived the study in collaboration with A.M.S, D.I.B. and I.H. N.B.K. designed and ran the models. A.M.S. constructed the velocity data sets, K.D.M. calculated the surface melt water contribution. F.G.-C. provided Elmer/Ice outputs. J.A.M. provided internal and basal temperature maps. J.E.B. contributed to discussions of total mass balance. M.C. adapted an ice mask for the purposes of this study. S.H.L. assisted with error checking the code. W.T.C., R.S.F. and K.K.K. compiled mass budget information for comparison. N.J.K. assisted with figures. N.B.K. wrote the manuscript with input from all authors.

## Competing interests

The authors declare no competing interests.
