## [Peer Review File · Nature Communications]

REVIEWER COMMENTS

Reviewer #1 (Remarks to the Author):

"A First Constraint on Basal Melt-water Production of the Greenland Ice Sheet" by Karlsson, et al. provides the first comprehensive estimate of basal melting beneath the Greenland Ice Sheet. They combine estimates of basal melting from geothermal heat flux, basal friction flux, and latent heat from surface meltwater infiltrating to the bed. The analysis is performed for conditions corresponding approximately to present day and year 2000. The results show that basal melting represents a term that is a 5-10% contribution of the overall mass budget of the ice sheet, and it therefore important for accurate closure of the input-output mass change method. The results also demonstrate how the basal mass balance term is changing in time, albeit slowly.

Historically, it has been assumed that the basal mass balance term of the ice sheet mass budget was small and static and could safely be neglected. Until recently, the accuracy of the input-output mass budget technique probably made the approximation justifiable. However, the increase in sophistication of ice sheet mass balance methods makes the current study timely and a significant contribution to understanding current and future changes in the Greenland Ice Sheet. The methodology for the approximation and uncertainty of the geothermal flux term and the surface meltwater term is sound and appropriate. The inclusion of the surface meltwater term is an important addition that has traditionally been ignored but shown here and by previous work by the authors to be important. However, the method used for the basal friction term is outdated and of questionable accuracy. As that term is generally the largest contribution to the overall basal melting calculation, this approach introduces unknown uncertainty to the results. This concern is elaborated on in the comments below.

**** Major concern ****

The major limitation of the study is the use of significant simplifications in stress balance for the calculation of the frictional heating term. The study uses an assumption of the shallow ice approximation for calculation of the basal sliding velocity and basal shear stress. However, despite being a commonly used simplification historically, the shallow ice approximation is only theoretically valid where basal sliding is small, which is precisely where basal frictional heating is small or absent. When sliding is present, the amount of deformation in the ice column often deviates largely from that predicted by the shallow ice velocity (see for example, Ryser, C., Lüthi, M.P., Andrews, L.C., Hoffman, M.J., Catania, G.A., Hawley, R.L., Neumann, T.A., Kristensen, S.S., 2014. Sustained high basal motion of the Greenland ice sheet revealed by borehole deformation. *J. Glaciol.* 60, 647–660. doi:10.3189/2014JoG13J196). This means the sliding velocity predicted by the currently used method may be too low. At the same time, the basal shear stress in regions of high ice velocity is often far below the driving stress, which violates the shallow-ice assumption employed by the current approach that the two stresses are equal. While these two errors are compensating, it cannot be expected that they will balance out. Furthermore, because of that assumption, the trends in basal frictional melting in Figure 3 are entirely due to changing velocity, while the basal shear stress contribution to changing basal friction is unaccounted for. Despite these issues, the error introduced by the low-order stress balance is not considered in the error budget. The use of a higher-order stress approximation would eliminate these concerns. It would also resolve basal friction at small scales and in small outlet glaciers, which may be a significant contribution to the total that is currently ignored.

At present, more accurate methods are widely available in the community. As the basal friction term is responsible for over half the total basal melting, a more accurate approach for this term is warranted. Most major ice sheet models now have existing model output for basal sliding and basal shear stress that could be used directly for the basal friction calculation required by this study. Ideally such a calculation would be done with a Stokes or 3d higher-order (first-order) flow model. Models with formal optimization methods could be used to calculate the basal friction flux separately for each of the years considered in the study. Some ice sheet models (MALI, ISSM, likely others) have the ability to solve the ice sheet temperature field simultaneously, reducing or eliminating the need for

estimating englacial temperatures from radar attenuation data and the associated large uncertainties. It may turn out that the simpler approximation currently used in the study results in similar results to a more accurate higher-order stress balance calculation, but at present there is no way to know the error introduced by the current approach. A valuable and interesting side product of improving the method used in the paper would be an assessment if the shallow ice approach currently used here is in fact adequate for large scale studies such as this. Use of a higher-order model estimate of basal friction flux would also allow the possibility of calculating the magnitude of substantially altered basal friction during summer, which could potentially impact of the annual basal melting budget.

**** Other general comments ****

1. There is one component of basal mass balance missing, which is melting due to channels within the subglacial drainage system. This is essentially a form of the latent heat term already considered where there is a positive feedback due to the gravitational potential energy included in new water being added to the system. There is no existing observational or theoretical evidence to suggest that this term is significant, but for completeness it should be mentioned. Estimating it would require a validated ice-sheet wide subglacial hydrology model, which I think is well beyond the scope of this paper.

2. Is the McGregor mask of frozen/thawed basal thermal state used for all three terms or just the geothermal term. I would expect the places where the other two terms are significant to be entirely within the thawed region of the mask, and indeed it appears that is the case through visual inspection. For consistency, it would be useful if this could be confirmed and reported on.

**** Specific comments ****

16, 18, 19: It is not generally obvious what would cause a change in basal melt for the ice sheet. The abstract should mention the mechanism by which this is occurring.

39-40: Also reference:

Slater, D.A., Felikson, D., Straneo, F., Goelzer, H., Little, C.M., Morlighem, M., Fettweis, X., Nowicki, S., 2020. Twenty-first century ocean forcing of the Greenland ice sheet for modelling of sea level contribution. *Cryosph.* 14, 985–1008. doi:10.5194/tc-14-985-2020

Rignot, E., Xu, Y., Menemenlis, D., Mougnot, J., Scheuchl, B., Li, X., Morlighem, M., Seroussi, H., den Broeke, M. van, Fenty, I., Cai, C., An, L., Fleurian, B. de, 2016. Modeling of ocean-induced ice melt rates of five west Greenland glaciers over the past two decades. *Geophys. Res. Lett.* 43, 6374–6382. doi:10.1002/2016GL068784

66: This uncertainty notation is potentially confusing. I recommend explaining it with words in parentheses at first usage.

72: Why ignore the uncertainties between datasets if you have already calculated them?

122: Missing)

113-132: The Results section for basal friction included some results for changes over time, but this section for surface-melt contribution does not.

134: "basal discharge" here is potentially misleading, as basal discharge in reality will be swamped by contribution from surface melt draining to the bed and being transported by the subglacial drainage system. "basal melt discharge" or similar would be better.

134: comma needed after "year"

138-9: Can you put uncertainties on these percentages?

144: Remind the reader what year(s) the reference period is here. When I read this sentence, I was initially wondering how this was different than the quantity discussed in the previous paragraph.

173: See comment for line 134 about "basal discharge".

177: I have low confidence in this conclusion based on the method of calculating basal friction melt.

182: See comment for line 134 about "basal discharge".

213-214: It is unclear what is being communicated about BedMachine v2 here.

230: "through" -> "from"

242-244: This is confusing. Why does the first sentence say 3 datasets are used if the second sentence says one of them is not used? On line 64, it says 3 datasets are used.

283: The statement that "All other uncertainties are likely of secondary importance" is overly speculative. It is very possible that the simplified stress balance approximation introduces greater uncertainty.

285: I do not know where in this reference this method is stated to be first order. Formally the shallow ice approximation is a zero-order approximation of the stress balance. See, for example: Kirchner, N., Hutter, K., Jakobsson, M., Gyllencreutz, R., 2011. Capabilities and limitations of numerical ice sheet models: a discussion for Earth-scientists and modelers. *Quat. Sci. Rev.* 30, 3691–3704. doi:10.1016/j.quascirev.2011.09.012

Fig 1: The description of shaded black and grey areas should be in the description of panel A, not B. The caption should describe the blue contours in panel E.

Fig.2 : See comment for line 134 about "basal discharge".

Reviewer #2 (Remarks to the Author):

Karlsson et al.

Summary:

The manuscript presents the first ice-sheet estimates of the contribution of basal melting to Greenland mass loss. Three sources of basal heat were considered: geothermal heat, frictional heat, and viscous heat dissipation from surface runoff. Using a combination of geophysical-, remote sensing observations, and a regional climate model, the authors calculated the changes of basal heat content from frictional heat since 2000 and from viscous heat dissipation since 1960. Their results demonstrated that frictional heating constitutes the majority of the present-day basal melting. Over the last two decades, increased viscous heat dissipation from surface runoff has the highest contribution toward the basal heat budget. The authors projected that given the current trajectory of increasing surface melting and glacier flow acceleration, basal melting would play an increasingly more significant role in the mass losses from the Greenland ice sheet.

Overall, the manuscript is easily readable, and it presents some exciting results on an understudied component of the Greenland mass loss. However, I have several concerns about how uncertainties are estimated for each of the heat components in this study:

1. Geothermal heat to basal melt conversion: The assumption that GHF in the “uncertain” regions labeled by MacGregor et al. (2015) would contribute to 50% of basal melting is questionable. In these uncertain zones, some models predict the basal temperature at the pressure melting while others predict a cold-based condition. The authors seem to be aware of these problems and thus presented two calculations: assuming the entire ice sheet is at pmp and assuming it is entirely frozen. However, they only show the lower bound values (e.g., “GHF component will increase by more than 70%”), and it is unclear whether they represent an ice-sheet averaged (and therefore further underestimating the impact of spatial differences between different GHF model and basal temperatures). I would suggest the authors either present the full uncertainty range or, better yet, include error maps for each of the heat component contributions.
2. Uncertainty in frictional heating: The authors did not explain the reasoning behind choosing a 5 °C variation in constant basal temperature offset to represent the uncertainty in their calculations. Also, the author’s approach in calculating stress balance is dependent on grid resolution; the authors did not discuss this impact of grid refinement on frictional heating uncertainty, beyond stating that their SIA approach cannot resolve individual ice streams and outlet glaciers. It is also unclear to me why publicly available MEaSURES summer surface velocities were not used. While summer velocity does not always have complete coverage over the entire ice sheet, in regions where the summer velocity exists, the authors should be able to use these observations to quantify better the uncertainty related to temporal changes in frictional heating.
3. Uncertainty in viscous heat dissipation: Inter-model differences in the surface runoff between MAR, RACMO, ECMWFd, and others can be quite considerable (10 – 30% differences) (e.g., Vernon et al., 2013). These differences should have a non-negligible impact on the spatial and temporal variations in viscous heat dissipation. Some form of sensitivity test on how these modeled differences in runoff within the study period would translate into uncertainty in viscous heating will be useful.

Specific comments:

L17: If the word count allows it, I would like to see a sentence or two summarizing the results on the three sources of basal melting (e.g., the most significant contributor to the recent changes).

L27: Add a comma before “corresponding.”

L28: Add a comma before “while.”

L44: You can be more specific about the time duration by stating, “over the last two decades.”

L57: Add a comma before “both.”

L70: What exactly do you mean by “local scale” (e.g., < 10 km, < 100 km, glacial catchment scale)?

L71: The 10% only represents an ice-sheet wide averaged value, and I imagine this value will be a lot greater in the SE region where the GHF estimates differ the most. Instead of presenting an averaged value, I would like to see a range that genuinely represents the uncertainty related to the GHF.

L94: While Greenland summer velocities are unavailable, several summer velocity products on a basin-scale do exist (<https://nsidc.org/data/NSIDC-0731/versions/1>; <https://nsidc.org/data/nsidc-0646>). Why don’t you use these products, at least on the large ice streams and the eight major outlets, to estimate how much differences these measurements will make?

L121-22: It is unclear what the word “This” is referring to. At first, I thought you meant that the 15%

uncertainty is translated to a 50% increase in basal melt changes. But upon the second read, I realized that "this" is most likely referring to the increase in Gt/year in the prior sentence. I would suggest either referring more precisely what "This" refers to or move this sentence ahead of the uncertainty sentence.

L123: Typo for "focussed."

L125: Add "due to surface meltwater" after "basal melt rates."

L137-39: The sentence beginning "At the ice sheet scale..." should be included in the abstract.

L147: comma before "while."

L149: How much of this stability is due to the fact you used winter velocities, which did not alter too much over the last 15 years, while most of the dynamic acceleration observed in the recent years mostly occurs in the summer melt season?

L155-56: The usage of "now larger" implies that some sectors have experienced a shift in the dominant basal melting mechanism. If that is the case, I believe they are interesting results that should be described more explicitly here.

L159-60: I am a bit surprised that a 170% increase in surface melt heating only resulted in a 6% increase in basal melt in the NO sector. What are the reasons behind this non-linearity, and is this specific to the NO sector?

L168: Should add "assuming subglacial water pressures at the ice overburden pressures." after "hydropotential." This assumption also doesn't hold near the ice sheet margin where fast, channelized type drainage develops. In regions like the SE, earlier studies have shown that subglacial drainage basins can evolve in response to changes in subglacial water pressures and that they do not always align with glaciological catchment (e.g., Lindbäck et al., 2015). How do uncertainties in the subglacial drainage basin influence your estimates of basal water fluxes?

L184-6: This sentence related to frictional heating is highly oversimplified and does not capture, e.g., the complexity related to how subglacial drainage morphology may evolve in the future at the interior vs. close to the margin (e.g., Bartholomew et al., 2011), or how glacier shape influences the susceptibility of individual glaciers to dynamic thinning (e.g., Felikson et al., 2017), for example. There is space in the discussion to talk about these earlier ideas, which will help to put this manuscript results in a better context of the current scientific understanding.

L195: The word "disproportionally" implies you compare the effect of basal melting on the ice sheet and fjord processes to something. What is it, e.g., to other mechanisms that basal melting may impact or relative to the role of surface runoff role in influencing fjord conditions? Also, the use of "ice sheet process" is too generic; this paragraph is mainly about fjord process, so I would suggest considering rewriting this sentence along the lines of "Basal melting may ALSO have a large impact on fjord processes and ice-ocean interaction."

L200: Add a comma before "but."

L204: "Finally, recent and future increases in basal melting LIKELY HAVE a non-linear effect..."

References:

Bartholomew, I.D., P. Nienow, A. Sole, D. Mair, T. Cowton, M.A. King, and S. Palmer. 2011. "Seasonal Variations in Greenland Ice Sheet Motion: Inland Extent and Behaviour at Higher Elevations." *Earth*

and Planetary Science Letters 307 (3–4): 271–78. <https://doi.org/10.1016/j.epsl.2011.04.014>.

Felikson, Denis, Timothy C. Bartholomaus, Ginny A. Catania, Niels J. Korsgaard, Kurt H. Kjær, Mathieu Morlighem, Brice Noël, et al. 2017. "Inland Thinning on the Greenland Ice Sheet Controlled by Outlet Glacier Geometry." *Nature Geoscience* 10 (5): 366–69. <https://doi.org/10.1038/ngeo2934>

Lindbäck, K, R Pettersson, A L Hubbard, S H Doyle, D. van As, A B Mikkelsen, and A. A Fitzpatrick. 2015. "Subglacial Water Drainage, Storage, and Piracy beneath the Greenland Ice Sheet." *Geophysical Research Letters* 42 (18): 7606–14. <https://doi.org/10.1002/2015GL065393>.

MacGregor, Joseph A., Mark A. Fahnestock, Ginny A. Catania, Andy Aschwanden, Gary D. Clow, William T. Colgan, S. Prasad Gogineni, et al. 2016. "A Synthesis of the Basal Thermal State of the Greenland Ice Sheet." *Journal of Geophysical Research: Earth Surface* 121 (7): 1328–50. <https://doi.org/10.1002/2015JF003803>.

Vernon, C. L., J. L. Bamber, J. E. Box, M. R. van den Broeke, X. Fettweis, E. Hanna, and P. Huybrechts. 2013. "Surface Mass Balance Model Intercomparison for the Greenland Ice Sheet." *The Cryosphere* 7 (2): 599–614. <https://doi.org/10.5194/tc-7-599-2013>.

Replies to review of: “A First Constraint on Basal Meltwater Production of the Greenland Ice Sheet” by Karlsson et al.

Dear reviewers,

Please find below our point-by-point responses. We have indicated reviewers' comments in **blue type** and our responses in black. Quotes from the updated manuscript are in *italics*. In brief, the main changes to the manuscript are as follows:

1. The way the uncertainties are handled for the geothermal flux has been changed in order to incorporate the full range of possible values.
2. The basal melt stemming from friction heat is estimated using Elmer/Ice, a Full Stokes ice-flow model. Fabien Gillet-Chaulet has provided outputs from Elmer/Ice and is now a co-author.
3. In order to obtain a more robust estimate of surface melt-water volumes, we use the mean value from 13 regional climate models (recently published as part of a surface mass balance intercomparison project).

Thank you for all the work you put into this manuscript and our sincere thanks for the constructive and helpful feedback and comments.

Best,

Nanna B. Karlsson on behalf of all coauthors

Reviewer #1 (Remarks to the Author):

A First Constraint on Basal Melt-water Production of the Greenland Ice Sheet” by Karlsson, et al. provides the first comprehensive estimate of basal melting beneath the Greenland Ice Sheet. They combine estimates of basal melting from geothermal heat flux, basal friction flux, and latent heat from surface meltwater infiltrating to the bed. The analysis is performed for conditions corresponding approximately to present day and year 2000. The results show that basal melting represents a term that is a 5-10% contribution of the overall mass budget of the ice sheet, and it therefore important for accurate closure of the input-output mass change method. The results also demonstrate how the basal mass balance term is changing in time, albeit slowly.

Historically, it has been assumed that the basal mass balance term of the ice sheet mass budget was small and static and could safely be neglected. Until recently, the accuracy of the input-output mass budget technique probably made the approximation justifiable. However, the increase in sophistication of ice sheet mass balance methods makes the current study timely and a significant contribution to understanding current and future changes in the Greenland Ice Sheet. The methodology for the approximation and uncertainty of the geothermal flux term and the surface meltwater term is sound and appropriate. The inclusion of the surface meltwater term is an important addition that has traditionally been ignored but shown here and by previous work by the authors to be important. However, the method used for the basal friction term is outdated and of questionable accuracy. As that term is generally the largest contribution to the overall basal melting calculation, this approach introduces unknown uncertainty to the results. This concern is elaborated on in the comments below.

**** Major concern ****

The major limitation of the study is the use of significant simplifications in stress balance for the calculation of the frictional heating term. The study uses an assumption of the shallow ice approximation for calculation of the basal sliding velocity and basal shear stress. However, despite being a commonly used simplification historically, the shallow ice approximation is only theoretically valid where basal sliding is small, which is precisely where basal frictional heating is small or absent. When sliding is present, the amount of deformation in the ice column often deviates largely from that predicted by the shallow ice velocity (see for example, Ryser, C., Lüthi, M.P., Andrews, L.C., Hoffman, M.J., Catania, G.A., Hawley, R.L., Neumann, T.A., Kristensen, S.S., 2014. Sustained high basal motion of the Greenland ice sheet revealed by borehole deformation. *J. Glaciol.* 60, 647–660. doi:10.3189/2014JoG13J196). This means the sliding velocity predicted by the currently used method may be too low. At the same time, the basal shear stress in regions of high ice velocity is often far below the driving stress, which violates the shallow-ice assumption employed by the current approach that the two stresses are equal. While these two errors are compensating, it cannot be expected that they will balance out. Furthermore, because of that assumption, the trends in basal frictional melting in Figure 3 are entirely due to changing velocity, while the basal shear stress contribution to changing basal friction is unaccounted for. Despite these issues, the error introduced by the low-order stress balance is not considered in the error budget. The use of a higher-order stress approximation would eliminate these concerns. It would also resolve basal friction at small scales and in small outlet glaciers, which may be a significant contribution to the total that is currently ignored.

At present, more accurate methods are widely available in the community. As the basal friction term is responsible for over half the total basal melting, a more accurate approach for this term is warranted. Most major ice sheet models now have existing model output for basal sliding and basal shear stress that could be used directly for the basal friction calculation required by this study. Ideally such a calculation would be done with a Stokes or 3d higher-order (first-order) flow model. Models with formal optimization methods

could be used to calculate the basal friction flux separately for each of the years considered in the study. Some ice sheet models (MALI, ISSM, likely others) have the ability to solve the ice sheet temperature field simultaneously, reducing or eliminating the need for estimating englacial temperatures from radar attenuation data and the associated large uncertainties. It may turn out that the simpler approximation currently used in the study results in similar results to a more accurate higher-order stress balance calculation, but at present there is no way to know the error introduced by the current approach. A valuable and interesting side product of improving the method used in the paper would be an assessment if the shallow ice approach currently used here is in fact adequate for large scale studies such as this. Use of a higher-order model estimate of basal friction flux would also allow the possibility of calculating the magnitude of substantially altered basal friction during summer, which could potentially impact of the annual basal melting budget.

Reply:

The reviewer brings up three points: 1) that the errors associated with the shallow-ice approximation are unknown and that a more accurate approach for obtaining the basal friction term is warranted, 2) that the basal shear stress contribution to changing basal friction is unaccounted for, and 3) that internal temperatures should be sourced from ice-sheet models rather than derived from radar-attenuation data. We address the concerns below (but see also response to reviewer #2 regarding inclusion of more velocity datasets).

1. In order to obtain a more accurate estimate of basal friction, we now utilise output from a simulation conducted by the Elmer/Ice model. This is a Full Stokes ice-flow model (i.e. it includes all stresses) and the output has been optimised to fit observations of surface velocity (using a multi-annual velocity dataset spanning 1995-2015). The addition of the Elmer/Ice results allow us to resolve the friction term on a 1km grid substantially improving the accuracy of our estimates. Due to the computational efforts needed to produce these results, we continue to use the simplified stress-balance for investigating the temporal evolution of the friction term. However, we are now able to quantify the uncertainty related to our use of the shallow-ice approximation by directly comparing with the Elmer/Ice outputs. We find that the total basal melt rate from the shallow-ice approximation is 31% higher than those derived from the Elmer/Ice results.

2. Yes, it is correct that the increase in basal melt from the friction heating is solely due to increased surface velocities. We based all our calculations on winter velocities where the subglacial system is the least active in order to capture annual/decadal increases in basal melt that are independent of the summer dynamics of individual years. Our point is that as slow-flowing areas speed up in response to warmer climate, the friction heat will also increase. The reviewer also notes that "Use of a higher-order model estimate of basal friction flux would also allow the possibility of calculating the magnitude of substantially altered basal friction during summer". We also address this comment in our response to reviewer #2 but we note here that using a higher-order model to capture seasonal changes in basal friction due to, for example, summer speed ups, would require model inversion for each of those velocity maps. That is computationally expensive and beyond the scope of this paper's aim which is to provide a first constraint on basal melt rates. Furthermore, if the velocity maps do not cover the whole ice sheet, unphysical velocity contrasts appear that are problematic for the basal sliding calculation. In order to quantify the likely underestimation of the friction heat due to the use of winter velocities, we have included two complete summer velocity maps (from summers 2018 and 2019). The resulting basal melt rates are 5% higher for these summer maps and we now include this 5% discrepancy in our total uncertainty assessment.

3. We respectfully disagree that temperatures generated by ice-sheet models are better or more accurate than those derived from radar attenuation. As shown in MacGregor et al., 2016 there is a large discrepancy between ice-flow models when it comes to temperature, leading to significant disagreements in basal and internal ice temperatures. Furthermore, most higher-order models compute ice temperatures using one of two approaches; they compute a steady state temperature, or they make use of an external temperature field to capture the long-term memory of the temperature field. In the former case, the temperature field only reflects present-day temperatures and accumulation rates thus deviating from the true present state of the ice sheet. In the latter case, lower-order models typically generate the external temperature field. This is necessary in order to be able to run the model for the required 10s-100s of thousand years. The “paleo temperature field” thus includes the thermal memory of the ice sheet but is influenced by uncertainties in the past air temperatures and accumulation rates that are needed to drive the models.

Actions taken:

Manuscript section on Results (Frictional heat contribution to basal melt) now presents the new estimates from the Elmer/Ice model. The Methods section has been updated to discuss the uncertainties relating to the Elmer/Ice model. The discussion of uncertainties relating to the shallow-ice approximation has also been expanded including reference to the Ryser et al., (2014) paper. The temporal change in friction heat is still based on the shallow-ice approximation but with updated uncertainties. All values have been updated in Table 1 and throughout the manuscript.

Figure 3 has been updated with the addition of two summer velocity maps.

Supplementary material now contains direct comparison between basal melt rates from the two models using Elmer/Ice and the shallow-ice approximation, respectively.

**** Other general comments ****

1. There is one component of basal mass balance missing, which is melting due to channels within the subglacial drainage system. This is essentially a form of the latent heat term already considered where there is a positive feedback due to the gravitational potential energy included in new water being added to the system. There is no existing observational or theoretical evidence to suggest that this term is significant, but for completeness it should be mentioned. Estimating it would require a validated ice-sheet wide subglacial hydrology model, which I think is well beyond the scope of this paper.

We do consider the potential energy of the additional water generated as the meltwater flows towards the margin. As the reviewer notes, we cannot estimate where this water is generated as that would require a full subglacial hydrology model. Instead, we assume that the water is at the bed, and we track this additional water supply and its influence on the heat budget as the water travels through the subglacial system. We find that this term add less than 1% to the total meltwater output.

Actions taken:

The following has been added to the manuscript:

Line 333 (now line 428): *“We also keep track of the energy budget as meltwater enters the hydrological system and melts out channels thus producing additional meltwater. This additional meltwater in turn may melt out more channels in a positive feedback. Lacking information on the exact location of the channels, we assume that they are situated at the bed, and we calculate the potential energy of this additional melt. Locally, this leads to less than 1 % increase in basal melt rates. ”*

2. Is the McGregor mask of frozen/thawed basal thermal state used for all three terms or just the geothermal term. I would expect the places where the other two terms are significant to be entirely within the thawed region of the mask, and indeed it appears that is the case through visual inspection. For consistency, it would be useful if this could be confirmed and reported on.

The frozen/thawed mask is only used for the geothermal heat. We thank the reviewer for bringing this up and on reflection; we agree that for consistency this mask should be applied to the friction term too. We do not apply it to the surface melt-water heat because we have more faith in the spatial distribution of surface melt water than in the frozen/thawed mask.

Actions taken:

We now mask the friction-induced basal melt map (Figure 1E) with the estimate of basal conditions from MacGregor et al. (2016). Text has been updated accordingly in the Results and Methods section. All values have been updated in Table 1 and throughout the manuscript.

**** Specific comments ****

16, 18, 19: It is not generally obvious what would cause a change in basal melt for the ice sheet. The abstract should mention the mechanism by which this is occurring.

The abstract has been modified and now reads:

“As the Arctic warms, we anticipate that basal melt will continue to increase due to faster ice flow and more surface melting thus compounding current mass loss trends, enhancing solid ice discharge and modifying fjord circulation.”

39-40: Also reference:

Slater, D.A., Felikson, D., Straneo, F., Goelzer, H., Little, C.M., Morlighem, M., Fettweis, X., Nowicki, S., 2020. Twenty-first century ocean forcing of the Greenland ice sheet for modelling of sea level contribution. *Cryosph. 14*, 985–1008. doi:10.5194/tc-14-985-2020

Rignot, E., Xu, Y., Menemenlis, D., Mouginot, J., Scheuchl, B., Li, X., Morlighem, M., Seroussi, H., den Broeke, M. van, Fenty, I., Cai, C., An, L., Fleurian, B. de, 2016. Modeling of ocean-induced ice melt rates of five west Greenland glaciers over the past two decades. *Geophys. Res. Lett.* 43, 6374–6382. doi:10.1002/2016GL068784

Lines 39-40 (now lines 43-45) has been changed to:

“Subglacial discharge increases the total submarine melt flux [7,8] and plays an important role for Greenland outlet glaciers’ contribution to future sea-level rise[9,10].”

66: This uncertainty notation is potentially confusing. I recommend explaining it with words in parentheses at first usage.

Added to line 67 (now line 69): *“(note that our uncertainty range is asymmetrical and we use / to denote upper/lower range)”*

72: Why ignore the uncertainties between datasets if you have already calculated them?

In response to reviewer 2, we have changed the way we assess the uncertainty associated with the geothermal flux. As a consequence, this paragraph has been rewritten. The line now reads:

“We find that the difference in ice-sheet-wide basal melt between the geothermal datasets is <10%, however, by including the likely range of geothermal flux based on each dataset’s stated uncertainty, the final uncertainty range increases (see methods).”

122: Missing)

Added. Thanks.

113-132: The Results section for basal friction included some results for changes over time, but this section for surface-melt contribution does not.

The section about the surface melt water has been rewritten and changes over time are now in a separate section titled “Temporal evolution of frictional and surface melt-water heat”.

134: “basal discharge” here is potentially misleading, as basal discharge in reality will be swamped by contribution from surface melt draining to the bed and being transported by the subglacial drainage system. “basal melt discharge” or similar would be better.

Changed to “basal melt discharge” here and in other places in the manuscript.

134: comma needed after “year”

Added comma.

138-9: Can you put uncertainties on these percentages?

We do not think it is meaningful to add uncertainties for these percentages since neither the variables nor their errors are independent.

144: Remind the reader what year(s) the reference period is here. When I read this sentence, I was initially wondering how this was different than the quantity discussed in the previous paragraph.

This section has been rewritten and the changes in time of the basal melt are now discussed in a separate section titled “Temporal evolution of frictional and surface melt-water heat”.

173: See comment for line 134 about “basal discharge”.

Changed throughout the manuscript.

177: I have low confidence in this conclusion based on the method of calculating basal friction melt.

This section has been removed and we are no longer assessing the change in time for individual glacier catchments.

182: See comment for line 134 about “basal discharge”.

Changed throughout the manuscript.

213-214: It is unclear what is being communicated about BedMachine v2 here.

We have updated our calculation of surface melt-water volumes to use BedMachine v3 instead. This sentence has been deleted.

230: “through” -> “from”

Changed.

242-244: This is confusing. Why does the first sentence say 3 datasets are used if the second sentence says one of them is not used? On line 64, it says 3 datasets are used.

We only use two of the datasets for the southern tip of Greenland because the Fox Maule dataset has no coverage there. We have changed the sentence for clarity.

Actions taken:

Line 247-248 (now lines 273-275): “Note that one of the datasets (Fox Maule[17]) does not cover the southern tip of Greenland so in this region, the average geothermal flux map is based on only two datasets.”

283: The statement that “All other uncertainties are likely of secondary importance” is overly speculative. It is very possible that the simplified stress balance approximation introduces greater uncertainty.

This sentence has been removed. See reply to comment above and our reply to the main concern.

285: I do not know where in this reference this method is stated to be first order. Formally the shallow ice approximation is a zero-order approximation of the stress balance. See, for example: Kirchner, N., Hutter, K., Jakobsson, M., Gyllencreutz, R., 2011. Capabilities and limitations of numerical ice sheet models: a discussion for Earth-scientists and modelers. *Quat. Sci. Rev.* 30, 3691–3704. doi:10.1016/j.quascirev.2011.09.012

Thank you for this correction. We have changed the description of the shallow-ice approximation including the uncertainty estimate, and as consequence removed the sentence.

Fig 1: The description of shaded black and grey areas should be in the description of panel A, not B. The caption should describe the blue contours in panel E.

The figure caption has been updated.

Actions taken:

The caption now reads:

“Figure 1: (A) Mean geothermal flux from [21, 22, 23]. The shaded areas outline where bed conditions are likely frozen (black) or uncertain (gray) based on radar observations and numerical ice-flow models[24]. (B) Surface velocities from multi-year MEaSURES dataset[13]. (C) Heat generated by surface melt-water infiltration. (D) Basal melting from geothermal heating. Blue contours outline the 0 m per year extent. (E) Basal melting from frictional heating. Purple outlines show the glacial catchments of Sermeq Kujalleq, Kangerlussuaq and Helheim Glacier[50]. Blue contours outline the 10–2m per year extent. (F) Basal melting from surface water heating. Dashed gray contours outline the 2000 m above sea level elevation. (D), (E), and (F) have the same logarithmic scalebar.”

Fig.2 : See comment for line 134 about “basal discharge”.

Changed throughout the manuscript.

Reviewer #2 (Remarks to the Author):

Karlsson et al.

Summary:

The manuscript presents the first ice-sheet estimates of the contribution of basal melting to Greenland mass loss. Three sources of basal heat were considered: geothermal heat, frictional heat, and viscous heat dissipation from surface runoff. Using a combination of geophysical-, remote sensing observations, and a regional climate model, the authors calculated the changes of basal heat content from frictional heat since 2000 and from viscous heat dissipation since 1960. Their results demonstrated that frictional heating constitutes the majority of the present-day basal melting. Over the last two decades, increased viscous heat dissipation from surface runoff has the highest contribution toward the basal heat budget. The authors projected that given the current trajectory of increasing surface melting and glacier flow acceleration, basal melting would play an increasingly more significant role in the mass losses from the Greenland ice sheet.

Overall, the manuscript is easily readable, and it presents some exciting results on an understudied component of the Greenland mass loss. However, I have several concerns about how uncertainties are estimated for each of the heat components in this study:

Main concerns

1. Geothermal heat to basal melt conversion: The assumption that GHF in the “uncertain” regions labeled by MacGregor et al. (2015) would contribute to 50% of basal melting is questionable. In these uncertain zones, some models predict the basal temperature at the pressure melting while others predict a cold-based condition. The authors seem to be aware of these problems and thus presented two calculations: assuming the entire ice sheet is at pmp and assuming it is entirely frozen. However, they only show the lower bound values (e.g., “GHF component will increase by more than 70%”), and it is unclear whether they represent an ice-sheet averaged (and therefore further underestimating the impact of spatial differences between different GHF model and basal temperatures). I would suggest the authors either present the full uncertainty range or, better yet, include error maps for each of the heat component contributions.

We think there is a misunderstanding here regarding how we proceed in our calculation of the geothermal flux component. We agree that the available estimates of geothermal flux are uncertain, and in order to address this uncertainty we consider different scenarios including a high-end scenario where the entire ice sheet is at pressure melting point. We do not consider a scenario where the entire ice sheet is frozen. Our lower-end scenario entails that all “uncertain” areas are frozen: Lines 255-257 (in original manuscript): “Conversely, if we assume that all areas are frozen where observations are ambiguous (Fig. 1B, grey contour), the total geothermal melt component decreases by approximately 26 % (26 % - 27 %).” This is the lower bound that we include in our stated uncertainty, line 66-67 (in original manuscript): “Our estimate of total geothermal basal melt is 5.3+4.0/-1.4 Gt per year.”

We interpret this misunderstanding as a need for clarity in our approach and in our writing. We have therefore revised the way we calculate the GF uncertainties. We adapt the methodology by van Liefvering and Pattyn (2013) and consider the possible range of GF values. We then construct two end members: a cold state where we use the lower range of possible GF values and where “uncertain” areas are frozen, and a warm state where we use the upper range of possible GF values and where “uncertain” areas are thawed. We also include maps in the supplementary material showing the two end members and the resulting basal melt rates.

Actions taken:

New uncertainty estimate for the geothermal flux. Text updated in Results section and values in Table 1 updated. Methods section updated to describe the new approach. Supplementary material updated with information on the two end members.

2. Uncertainty in frictional heating: The authors did not explain the reasoning behind choosing a 5 °C variation in constant basal temperature offset to represent the uncertainty in their calculations. Also, the author's approach in calculating stress balance is dependent on grid resolution; the authors did not discuss this impact of grid refinement on frictional heating uncertainty, beyond stating that their SIA approach cannot resolve individual ice streams and outlet glaciers. It is also unclear to me why publicly available MEaSURES summer surface velocities were not used. While summer velocity does not always have complete coverage over the entire ice sheet, in regions where the summer velocity exists, the authors should be able to use these observations to quantify better the uncertainty related to temporal changes in frictional heating.

We use a 5°C variation based the agreement between internal temperatures and the frozen/thawed extent from MacGregor et al. (2016). With a temperature offset of 25°C, frozen areas (e.g., areas that are likely frozen according to models and observations) approach or reach melting point. Using a temperature offset of 15°C, thawed areas (e.g., areas that are likely at pressure melting point according to models and observations) are cold. Thus, going above a 5C range is not necessary. We have clarified this in the Methods section.

We have now included results from a Full Stokes ice-flow model on a 1km grid. This resolution resolves individual glacier outlets better. See also response to reviewer #1.

The MEaSURES dataset does not have complete coverage. Thus, we would have to focus on individual glacier outlets rather than ice-sheet scale estimates. This would introduce unphysical transitions in the velocity dataset that complicates the calculation of basal sliding. Furthermore, for individual glacier basins, the shallow-ice approximation becomes increasingly uncertain due to the coarse resolution. At the same time, it is not viable to perform an inversion with Elmer/Ice for every summer velocity data product. Instead, we have now included two summer velocity maps from 2018 and 2019 (Sentinel-1). The results show that summer basal melt is 5% higher than winter basal melt.

Actions taken:

As outlined in our response to reviewer 1, the results from the frictional heat term have now been revised and updated with the outputs from the Full Stokes model. In addition, we have added the following to the Methods section (Frictional heat):

Lines 331-338 (now lines 324-341): *“In order to investigate the uncertainties due to poorly constrained internal temperatures, we vary our constant temperature offset by $\pm 5^{\circ}\text{C}$. We chose $\pm 5^{\circ}\text{C}$ as a likely uncertainty range because comparison between the internal temperature and estimated basal conditions reveals that changing the offset by more than -5°C returns cold conditions in areas that are likely thawed at the bed[24], while changing the offset by more than $+5^{\circ}\text{C}$ returns warm conditions in areas that are likely frozen at the bed[24]. We find that a change in temperature of $\pm 5^{\circ}\text{C}$ leads to a change in basal melt from frictional heat by $\pm 25\%$ (for the 2018/2019 velocity dataset).”*

Lines 348-353 (now lines 351-356): *“We primarily make use of winter velocities potentially leading to an underestimation of annual basal melt rates since summer velocities are typically higher. We use winter*

velocities due to the lack of complete maps from summer observations. However, with the recent launch of Sentinel-1, it is possible to construct complete summer velocity maps, and we have included two maps from summers 2018 and 2019. The resulting basal melt rates are 5 % higher for these summer maps likely due to the increased ice-flow velocities.”

3. Uncertainty in viscous heat dissipation: Inter-model differences in the surface runoff between MAR, RACMO, ECMWFd, and others can be quite considerable (10 – 30% differences) (e.g., Vernon et al., 2013). These differences should have a non-negligible impact on the spatial and temporal variations in viscous heat dissipation. Some form of sensitivity test on how these modeled differences in runoff within the study period would translate into uncertainty in viscous heating will be useful.

In order to present a more robust estimate of the heat from surface melt water, we have updated our results and now use the mean of 13 different regional climate models that recently partook in a surface mass balance intercomparison project (Fettweis et al., 2020). Based on values in this publication, we set the uncertainty range to 30% using the inter-model differences. Note that the change in model also means a change in resolution (from 5km to 1km) and a change in the reported basal melt rates.

Actions taken:

The section Results (Surface-melt water heat contribution to basal melt) has been updated to reflect the new model input. Text and Table 1 have been updated with the new results. We have added a table (Table 2) showing the changes in basal melt rate over time due to changing surface melt-water volumes.

The first paragraph in section Methods (Subglacial water routing and viscous heat dissipation) now reads: *“We estimate the surface melt water contribution using previously published methodology[28] where heat estimates are derived from runoff values from the GrSMBMIP project (Greenland Surface Mass Balance Model Intercomparison Project). The GrSMBMIP project compiles results from 13 regional climate models during 1980-2012 CE and we use the average values from all 13 models. The reported spread in modelled surface melt water volumes is 30 % and we use this range as our uncertainty”*

Specific comments:

L17: If the word count allows it, I would like to see a sentence or two summarizing the results on the three sources of basal melting (e.g., the most significant contributor to the recent changes).

We have added the following sentence to the abstract: *“We find that the ice sheet's present basal melt production is 21.4 +4.4/-4.0 Gt per year, and that melt generated by basal friction is responsible for about half of this volume. We estimate that basal melting has increased by 2.9±5.2 Gt during the first decade of the 2000s.”*

L27: Add a comma before “corresponding.”

Comma added.

L28: Add a comma before “while.”

Comma added.

L44: You can be more specific about the time duration by stating, “over the last two decades.”

Sentence changed to: *“Here, we provide the first estimate of ice-sheet-scale basal melt and its recent change through the first decade of the 2000s.”*

L57: Add a comma before “both.”

This sentence has been changed and now reads: *“Although studies have found evidence of subglacial lakes [16, 17] and “units of disturbed radio-stratigraphy” [18, 19], associated volumes are negligible in the context considered here.”*

L70: What exactly do you mean by “local scale” (e.g., < 10 km, < 100 km, glacial catchment scale)?

This sentence has been removed.

L71: The 10% only represents an ice-sheet wide averaged value, and I imagine this value will be a lot greater in the SE region where the GHF estimates differ the most. Instead of presenting an averaged value, I would like to see a range that genuinely represents the uncertainty related to the GHF.

See reply above regarding the reviewer’s point 1, the treatment of uncertainties for the GF term.

L94: While Greenland summer velocities are unavailable, several summer velocity products on a basin-scale do exist (<https://nsidc.org/data/NSIDC-0731/versions/1>; <https://nsidc.org/data/nsidc-0646>). Why don’t you use these products, at least on the large ice streams and the eight major outlets, to estimate how much differences these measurements will make?

See reply above regarding the reviewer’s point 2 – uncertainty of the frictional heat term.

L121-22: It is unclear what the word “This” is referring to. At first, I thought you meant that the 15% uncertainty is translated to a 50% increase in basal melt changes. But upon the second read, I realized that “this” is most likely referring to the increase in Gt/year in the prior sentence. I would suggest either referring more precisely what “This” refers to or move this sentence ahead of the uncertainty sentence.

This sentence was removed when the section was rewritten.

L123: Typo for “focussed.”

“Focussed” is allowed spelling in British English.

L125: Add “due to surface meltwater” after “basal melt rates.”

Changed.

L137-39: The sentence beginning “At the ice sheet scale...” should be included in the abstract.

Thank you. We have added a sentence to the abstract stating that half of the basal melt is due to friction.

L147: comma before “while.”

Comma added.

L149: How much of this stability is due to the fact you used winter velocities, which did not alter too much over the last 15 years, while most of the dynamic acceleration observed in the recent years mostly occurs in the summer melt season?

This sentence was removed when the section was rewritten. Given the uncertainties associated with the shallow-ice approximation, we have decided against interpreting on the change (or lack thereof) in increased friction melt.

L155-56: The usage of “now larger” implies that some sectors have experienced a shift in the dominant basal melting mechanism. If that is the case, I believe they are interesting results that should be described more explicitly here.

Thank you for this suggestion. We have added the following sentence (in lines 196-198): *“In the NE, NO and SW sectors, the basal melt rates from 2012 surface melt water exceed the baseline friction-melt term implying a shift in principal basal melting process.”*

L159-60: I am a bit surprised that a 170% increase in surface melt heating only resulted in a 6% increase in basal melt in the NO sector. What are the reasons behind this non-linearity, and is this specific to the NO sector?

This sentence was removed when the section was revised but the small increase is due to the relatively low surface melt-water volumes in the NO sector.

L168: Should add “assuming subglacial water pressures at the ice overburden pressures.” after “hydropotential.” This assumption also doesn’t hold near the ice sheet margin where fast, channelized type drainage develops. In regions like the SE, earlier studies have shown that subglacial drainage basins can evolve in response to changes in subglacial water pressures and that they do not always align with glaciological catchment (e.g., Lindbäck et al., 2015). How do uncertainties in the subglacial drainage basin influence your estimates of basal water fluxes?

The uncertainties in the subglacial drainage basins do not affect our estimates of basal mass loss. For the mass loss, we are primarily concerned with the loss of mass in each grid cell and less with the eventual exit of the basal water. For our estimates of basal water fluxes, the uncertainty in subglacial drainage basins may influence fluxes into individual fjords for basins that are not controlled by bed topography (like the area presented in the Lindbäck 2015 study). That is not the case for the glaciers presented here, Sermeq Kujalleq, Kangerlussuaq and Helheim Glaciers all have well-defined troughs that likely will guide the subglacial water (channelized or not).

Actions taken:

Changed the sentence to (lines 156-157): *“Here, we calculate the individual subglacial basins using the hydropotential assuming that the subglacial water pressure is at ice overburden pressure[31]”*

L184-6: This sentence related to frictional heating is highly oversimplified and does not capture, e.g., the complexity related to how subglacial drainage morphology may evolve in the future at the interior vs. close to the margin (e.g., Bartholomew et al., 2011), or how glacier shape influences the susceptibility of individual glaciers to dynamic thinning (e.g., Felikson et al., 2017), for example. There is space in the discussion to talk about these earlier ideas, which will help to put this manuscript results in a better context of the current scientific understanding.

We agree that the sentence was oversimplified and have now modified it as outlined below.

Actions taken:

Paragraph changed to: *“Basal melt will change as the Greenland ice sheet responds to a warming climate. The frictional heat will increase if the areal extent of the fast-flowing regions expand, leading to an increase basal melt production. However, the impact of climate change in ice-stream dynamics is complex and thus, we cannot predict by how much the friction term will increase. Based on the recent past (Fig. 3), if glaciers*

continue to accelerate, basal melt water production may increase by ~0.1 Gt every year into the foreseeable future.”

L195: The word “disproportionally” implies you compare the effect of basal melting on the ice sheet and fjord processes to something. What is it, e.g., to other mechanisms that basal melting may impact or relative to the role of surface runoff role in influencing fjord conditions? Also, the use of “ice sheet process” is too generic; this paragraph is mainly about fjord process, so I would suggest considering rewriting this sentence along the lines of “Basal melting may ALSO have a large impact on fjord processes and ice-ocean interaction.”.

Changed as suggested. Thank you.

L200: Add a comma before “but.”

Comma added.

L204: “Finally, recent and future increases in basal melting LIKELY HAVE a non-linear effect...”

Changed as suggested.

References

Fettweis, X. et al., GrSMBMIP: intercomparison of the modelled 1980–2012 surface mass balance over the Greenland Ice Sheet. *The Cryosphere* 14, 3935–3958 (2020).

<https://tc.copernicus.org/articles/14/3935/2020/>

MacGregor, J. A., et al., A synthesis of the basal thermal state of the Greenland Ice Sheet. *Journal of Geophysical Research: Earth Surface* 121, 1328–1350 (2016).

Van Liefferinge, B. & Pattyn, F. Using ice-flow models to evaluate potential sites of million year-old ice in Antarctica. *Climate of the Past* 9, 2335–2345 (2013).

REVIEWER COMMENTS

Reviewer #1 (Remarks to the Author):

The manuscript "A First Constraint on Basal Melt-water Production of the Greenland Ice Sheet" by Karlsson, et al. has been revised to address reviewer comments. I appreciate the thorough response to reviewers made by the authors. The addition of model output from Elmer/Ice for calculating the basal friction flux addresses my primary concern with the original manuscript. While it is unfortunate that additional model inversions were not feasible to evaluate the change in basal friction melt over time, the computing constraints are understandable. Also, the simplified stress balance analysis shows that the changes in time due to this process are not the dominant cause of changes in basal melt over time. Thus, I think the level of approximation used is reasonable. While this section has been greatly clarified by the revisions, the revisions have added confusion for me in the section describing the viscous heat dissipation. The methods for this section require further clarification for a reader to understand what was done. These clarifications will not change the manuscript's findings, but are essential for documentation and reproducibility. I also have a handful of minor comments to improve clarity elsewhere in the manuscript.

Questions about viscous heat methods:

419: It is critical that it is explicitly stated that this definition of the hydropotential assumes the ice is always at the floatation point everywhere. Eq. 7 is not a general hydropotential equation.

423-6: How is V determined? I'm guessing it is some sort of routing algorithm, but there is no information about what that is.

431-5: There is no information about how it is calculated how and where channels melt out. I can only imagine this requiring a full subglacial hydrology model, but none is described.

Other issues:

46: You could also add a sentence or two about the role of subglacial melt in affecting the presence and magnitude of basal slip. Basal melt also plays a role in the seasonal cycle of subglacial hydrology.

51-52: Should also now mention output from an ice sheet model

54: average *annual* surface melt-water

78: Do you mean "unevenly" distributed? The rest of the paragraph discusses spatial variability.

85-90: I realize there is more detail in the Methods, but it would help the reader to add in two more pieces of information here: the model grid resolution and where the model ice temperature comes from.

183-4: Perhaps the biggest uncertainty is the assumption that basal shear stress remains constant when velocity changes, which is almost never the case. This should be acknowledged here. Ignoring this effect will generally overestimate your basal friction calculations.

226: It is unknown if basal melt is *the* primary source of winter subglacial discharge. There can be long lags in the subglacial drainage system due to storage, and delayed release of summer surface melt reaching the bed may indeed be the largest source in winter subglacial discharge. I recommend changing "the primary source" to "a primary source".

243: The simplified stress-balance model is referenced here before it is introduced below. Same with

line 252.

307: The new analysis using Elmer to calculate basal friction is a very strong addition to the manuscript and provides much more confidence about this basal melting term. However, the relation between the Elmer method and the SIA method is a little hard to follow. I suggest adding subsection headings within the Frictional Heat section for something like "Full stress balance", "Simplified stress balance", and "Changes in time", etc.

343: The -/+ symbol here is inconsistent with the +/- symbols used elsewhere.

356-8: It should be explicitly stated here that while the velocity was changed, the basal shear stress was not, which leads to what is likely an overestimate of the melt rate.

359: Increased velocities of summer occur over more like 25% of the year in Greenland.

391-401: It should be acknowledged here that when exploring changes in velocity over time, the basal shear stress is left unchanged (again leading to a likely overestimate of melt rate).

438: Data should be made available prior to publication.

Reviewer #2 (Remarks to the Author):

Karlsson et al.,

Overall, the authors have done a great job addressing some of my earlier concerns regarding the uncertainty in GHF. The inclusion of a full-Stokes model and 13 regional climate models are great improvements and frankly, it is the best we can do with our current modeling and observational capabilities. I only have stylistic suggestions and some nitpicking, minor comments to add.

Line comments:

L27: The main difference between altimetry and gravimetry is that GRACE does not require any assumptions about firn densification etc. in the height-to-area-to-volume conversion; GRACE directly measure mass. So I would suggest rewording this sentence to highlight that more, something like "directly" measuring "mass" changes "using" "gravimetry".

L30-31: Stylistic suggestion: the average mass balance of the ice sheet "between 2005-2015" is ... And then you can get rid of the use of parentheses in the end.

L35: Nitpicking but you can be more specific here and say: accurate representation of the "climatic and dynamic" mass loss terms.

L49: Add a comma before "while"

L51: I'd specify that "surface melt input to the bed" as opposed to surface melt production that matters to heat transfer to the basal melt component.

L52: Remove "the" after quantify.

L53: Remove "as well as"

L57: Add "or land-margin" since you considered both marine and land-terminating glaciers here.

L58: I'm not sure there is an absolutely 0% chance that large subglacial lakes exist in Greenland (e.g. Oswald & Gogineni, 2012 seems to think there are plenty in the ice sheet interior), but I think it is likely that there is a limited "long-term" storage of meltwater in subglacial lakes given the high input of surface meltwater fluxes and geometric effects of the ice sheet that you have highlighted. So may I suggest we reword this sentence something along the lines: preclude the existence of "long-term meltwater storage in" large subglacial lakes?

L78: Is it "evenly distributed"? Looking at Fig. 1D, I'd say geothermal basal melt is quite spatially variable between the ice sheet interior and lower elevation regions.

L85: Remove "The" before frictional heat. And same for the next sentence before frictional heat.

L92: Stylistic suggestion: Note that Elmer/Ice predicts...

L99: Remove "the" before geothermal flux instead say "geothermal heat fluxES".

L103: Stylistic suggestion: replace ", and" with "with"

L107: Add more information here by saying "... important in slow-moving sectors in areas that on average move at XXX myr⁻¹ or less".

L111: largest "and fastest"

L115: Remove "the" before "heat"

L119: Instead of using a single 2000 m cutoff for all of Greenland, why didn't you use the ELA to capture the impact of spatial difference in the ablation zone area? (assuming we only have top-to-bottom water transport in the ablation zone)

L131: Remove "the" before melt

L132: Remove "the" before conduits

L172: Mention it's an underestimation because you are not using summer surface velocity.

L175: Any thoughts as to why the Full-Stokes model gives a lower estimate than the simplistic approach, especially in NE and NW sectors? Is it because Elmer/Ice is missing some complicated feedback mechanisms or something else?

L179: remove "the" after "include" and "the" before "basal shear layer" and turn "uncertainty" into plural since there are multiple sources of uncertainty in velocity.

L220: Again, I believe there is a distinction between increased surface melt production vs melt input to the bed here. While melt production will likely increase under warming climate scenarios with both increasing melting in the ablation zone and migration of surface ablation to higher elevations, we may also see an increase in surface runoff, e.g. if meltwater refreezes in firn forming a low permeability barrier that reduces vertical percolation (e.g. MacFerrin et al., 2019). So I'd caution putting in a broad-stroke statement like this, and maybe reword this to something more like "heat transported by surface meltwater will increase with greater meltwater production, which will likely increase meltwater delivery to the bed especially in the ablation zone."

Oswald, G. K. A., & Gogineni, S. P. (2012). Mapping basal melt under the northern Greenland ice

sheet. *IEEE Transactions on Geoscience and Remote Sensing*, 50(2), 585–592.
<https://doi.org/10.1109/TGRS.2011.2162072>

MacFerrin, M., Machguth, H., As, D. van, Charalampidis, C., Stevens, C. M., Heilig, A., et al. (2019). Rapid expansion of Greenland's low-permeability ice slabs. *Nature*, 573(7774), 403–407.
<https://doi.org/10.1038/s41586-019-1550-3>

Replies to review of: “A First Constraint on Basal Meltwater Production of the Greenland Ice Sheet” by Karlsson et al.

Dear reviewers,

We thank you for the detailed and thorough responses, and for your support of our work. We note that both reviewers appreciate our efforts to address their comments on the previous version of this manuscript. Please find below our point-by-point responses. We have indicated reviewers' comments in **blue type** and our responses in black. Quotes from the updated manuscript are in *italics*. Line numbers refer to the revised manuscript while numbers in parenthesis refer to the previous version.

We implemented all suggested changes, except the following two:

1. Reviewer 1 suggests that we change our definition of summer velocities (line 382-384 (359)). We prefer to keep the current definition in order to have a conservative upper bound for our uncertainty range.
2. Reviewer 1 notes that our simplified approach relies on the assumption that basal shear stress remains constant leading to an overestimate of basal friction. We agree that this is a key uncertainty but we disagree that the assumption always leads to an overestimate. We explain why in our detailed response below.

In addition to the reviewers' suggestion, we have changed figure 1E since we noticed that peripheral ice caps mistakenly were depicted as well. This has no impact on our results. We have also added the clarification that although we do not model basal freeze-on explicitly, the surface melt water heat term takes basal freeze-on into account as detailed in the original study by Mankoff and Tulaczyk (2017). Line 124-125 now reads: *“This entails that water is allowed refreeze locally due to supercooling as described in [34].”* In addition, a few sentences have been changed for clarity.

Our sincere thanks to both reviewers for their constructive and helpful feedback and comments.

Best,

Nanna B. Karlsson on behalf of all co-authors.

REVIEWER COMMENTS

Reviewer #1 (Remarks to the Author):

The manuscript “A First Constraint on Basal Melt-water Production of the Greenland Ice Sheet” by Karlsson, et al. has been revised to address reviewer comments. I appreciate the thorough response to reviewers made by the authors. The addition of model output from Elmer/Ice for calculating the basal friction flux addresses my primary concern with the original manuscript. While it is unfortunate that additional model inversions were not feasible to evaluate the change in basal friction melt over time, the computing constraints are understandable. Also, the simplified stress balance analysis shows that the changes in time due to this process are not the dominant cause of changes in basal melt over time. Thus, I think the level of approximation used is reasonable. While this section has been greatly clarified by the revisions, the revisions have added confusion for me in the section describing the viscous heat dissipation. The methods for this section require further clarification for a reader to understand what was done. These clarifications will not change the manuscript’s findings, but are essential for documentation and reproducibility. I also have a handful of minor comments to improve clarity elsewhere in the manuscript.

We thank the reviewer for their constructive comments, which we address in detail below.

Questions about viscous heat methods:

419: It is critical that it is explicitly stated that this definition of the hydropotential assumes the ice is always at the floatation point everywhere. Eq. 7 is not a general hydropotential equation.

The reviewer is correct that our assumption that water enters everywhere in the subglacial system is functionally the same as assuming floatation. We have clarified this sentence and it now reads:

Line 457-458 (419): *We assume that ice the subglacial water pressure is equal to the overburden pressure and that the subglacial water follows the steepest gradient of the hydropotential...*

423-6: How is V determined? I’m guessing it is some sort of routing algorithm, but there is no information about what that is.

The melt rate at the surface is obtained from the regional climate model. The water is then routed according to Eq. 7, i.e., following the hydropotential head. Information on the routing algorithm can also be found in Mankoff and Tulaczyk (2017). We have modified the sentences to include more details.

Line 465-468 (423-426): *[...] and c_p the specific heat of water $4184 \text{ J K}^{-1} \text{ kg}^{-1}$. We route the water using Eq. (7), assuming that water moves to the neighbouring cell with the lowest hydropotential. The routing algorithm is an industry-standard GIS (Geographic Information System) hydrological routing algorithm in GRASS GIS (Geographic Resource Analysis Support System GIS).*

431-5: There is no information about how it is calculated how and where channels melt out. I can only imagine this requiring a full subglacial hydrology model, but none is described.

We do not need a full subglacial model because we do not need to resolve individual conduits in order to keep track of the energy budget. We have rephrased the sentences for clarity:

Line 476-480 (431-435): *We also keep track of the energy budget as meltwater is routed through the hydrological system, producing additional meltwater. This additional meltwater in turn may melt out more water in a positive feedback. We do not resolve the location of individual conduits explicitly and thus lacking information on their exact location, we assume that they are situated at the bed, and we calculate the potential energy of this additional melt.*

Other issues

46: You could also add a sentence or two about the role of subglacial melt in affecting the presence and magnitude of basal slip. Basal melt also plays a role in the seasonal cycle of subglacial hydrology.

We have added the following sentences

Line 41-44 (46): *Secondly, the presence or absence of basal meltwater is important for the evolution of the subglacial system [7, 8], and recent studies have highlighted the importance of subglacial discharge for modifying the mass loss from marine-terminating glaciers [9, 10], it therefore plays an important role for Greenland outlet glaciers' contribution to future sea-level rise [11, 12].*

With citations of [7]: Magnússon, E., Björnsson, H., Rott, H., and Pálsson, F.: Reduced glacier sliding caused by persistent drainage from a subglacial lake, *The Cryosphere*, 4, 13–20, <https://doi.org/10.5194/tc-4-13-2010>, 2010. And [8]: Schoof, C. Ice-sheet acceleration driven by melt supply variability. *Nature* 468, 803–806 (2010). <https://doi.org/10.1038/nature09618>

51-52: Should also now mention output from an ice sheet model

Yes, thank you for spotting this omission. We have changed the sentence:

Line 51-53 (51-52): *We quantify the basal melt using estimates of geothermal flux, satellite-derived ice-surface velocities, surface and bed topographies, and outputs from an ice-sheet model and regional climate models.*

54: average *annual* surface melt-water

Changed to:

Line 54-55 (54): *[...] and average decadal/multi-decadal surface melt-water volumes from 1991-2012 [18].*

78: Do you mean “unevenly” distributed? The rest of the paragraph discusses spatial variability.

Yes, thank you. Changed as suggested.

85-90: I realize there is more detail in the Methods, but it would help the reader to add in two more pieces of information here: the model grid resolution and where the model ice temperature comes from.

We have added the following sentences to provide additional information:

Line 88-91 (85-90): *[...] where basal sliding and shear stress are related by a linear friction law [31]. Internal ice temperatures are obtained from a paleo spin-up run [33]. The model uses an anisotropic mesh where the horizontal resolution ranges from ~500 m to ~50 km, but here the original model results have been re-gridded on a 1 km equidistant grid.*

And in the Methods section, we now write:

Line 294-298: *Elmer/Ice uses an anisotropic mesh optimised to capture velocity and thickness variations, and insure a high resolution in the first 40 km from the ice-sheet edges. The resulting horizontal resolution ranges from ~500 m to ~50 km. Original model results have been re-gridded on a 1km equidistant grid for the post-processing [46]. The internal ice temperature field comes from a paleo spin-up of the SICOPOLIS model[33].*

183-4: Perhaps the biggest uncertainty is the assumption that basal shear stress remains constant when velocity changes, which is almost never the case. This should be acknowledged here. Ignoring this effect will generally overestimate your basal friction calculations.

Yes, this is indeed an inevitable uncertainty in the method, and we have added a comment to acknowledge it. However, we do not think it is too bad, since on the coarser scales at which this simplified model applies, the basal shear stress is essentially controlled geometrically. That is, it has to be such as to balance the driving stress, which has not changed much over the timescales considered. We agree that on smaller scales (1-2 ice thicknesses or less) the change in dissipation due to increased basal velocity depends on the correlation between velocity and shear stress. The basal velocity could speed up locally *because* of a decrease in basal stress, in which case assuming a constant basal stress would indeed give an overestimate of the local frictional heating. But that will result in an increase in basal stress elsewhere, where we have a corresponding underestimate (e.g. if sliding occurs according to a Weertman sliding law, the basal shear stress increases with increasing velocity, so treating the shear stress as fixed results in an underestimate of the frictional heating). However, on the coarser scales at which the simplified model is valid, it is reasonable to assume that the basal shear stress stays constant (for the purpose of calculating the frictional heating) because any changes must roughly average out over the coarser scale in order to ensure overall force balance. Finally, we note that our uncertainty range encompasses the uncertainty resulting from our assumption of constant basal stress. Therefore, the question of whether or not we overestimate does not influence our final results. We include the following sentence:

Line 202-204 (183-184): “[...] indicates that basal friction discharge has increased by $0.09 +0.04/-0.03$ Gt per year. Note that basal shear stress is assumed to remain constant. Over most of the ice sheet, glacier geometry (and hence driving stresses) did not change significantly during our study period, implying near-constant resisting stresses on the large spatial scale used in our simplified model.”

We have also added the following sentences in the paragraph describing the uncertainties:

Lines 193-196: *In particular, we assume that the basal shear stress remains constant despite the velocities changing (a reasonable approximation on the coarser scale of this approach, since overall force balance must be maintained, but which would not necessarily be true on a local scale).*

226: It is unknown if basal melt is **the** primary source of winter subglacial discharge. There can be long lags in the subglacial drainage system due to storage, and delayed release of summer surface melt reaching the bed may indeed be the largest source in winter subglacial discharge. I recommend changing “the primary source” to “a primary source”.

Changed as suggested.

243: The simplified stress-balance model is referenced here before it is introduced below. Same with line 252.

Yes, this section was intended as an introduction to the datasets that we use for the friction heat but we can see how this can be confusing. We have therefore merged the sections on “Surface and bed topographies” and “Ice velocity data” with the description of the two ice-flow models.

Changes to the manuscript: Merge of the two sections on data into the relevant method sections where the usage of the data is described.

307: The new analysis using Elmer to calculate basal friction is a very strong addition to the manuscript and provides much more confidence about this basal melting term. However, the relation between the Elmer method and the SIA method is a little hard to follow. I suggest adding subsection headings within the Frictional Heat section for something like “Full stress balance”, “Simplified stress balance”, and “Changes in time”, etc.

We are pleased that the reviewer finds that the addition of Elmer/Ice has strengthened our results. The section describing the ice flow models has been split into multiple sections to clarify our use of the models and our treatment of uncertainties. We now have separate sections on “Elmer/Ice”, “Shallow-ice approximation” and “Uncertainties” for the frictional heat term. We have also made use of additional spacing between paragraphs to delineate the different subsections.

Changes to the manuscript: New subsections introduced in the “Methods” section as described above.

343: The $-/+$ symbol here is inconsistent with the $+/-$ symbols used elsewhere.

The reason is that in this case, a change in $+5^{\circ}\text{C}$ leads to a change of -25% . We have written this out in the text to avoid confusion.

Line 364 – 367 (343): *We find that a change in temperature of $+5^{\circ}\text{C}$ leads to a change in basal melt from frictional heat by -25% , conversely a change in temperature of -5°C leads to a 25% increase in basal melt (for the 2018/2019 velocity dataset).*

356-8: It should be explicitly stated here that while the velocity was changed, the basal shear stress was not, which leads to what is likely an overestimate of the melt rate.

See reply above. We have added the following sentence:

Line 382-385 (356-358): *The resulting basal melt rates are 5% higher for these summer maps due to the increased ice-flow velocities. We note that in our simplified stress-approximation, an increase in surface velocity translates directly into an increase in friction heat because we assume that the resistance to sliding over the bed is constant regardless of velocity.*

359: Increased velocities of summer occur over more like 25% of the year in Greenland.

Yes, in general but there are regional variations where summer speed up occurs early in the season or continue well into the autumn. We state that “Assuming that summer velocities are representative for at most 50% of the year...” in order to have a conservative upper bound for our uncertainty. A shorter “assumed” summer would give us a lower uncertainty range.

No changes made to the manuscript here.

391-401: It should be acknowledged here that when exploring changes in velocity over time, the basal shear stress is left unchanged (again leading to a likely overestimate of melt rate).

See reply above. We have added the following sentence

Line 430-432 (391): *"Below, we outline the reasoning behind this conjecture. Again we note that our simplified stress-approximation assumes that the basal stress is constant."*

438: Data should be made available prior to publication.

Yes, data have now been assigned a DOI and are available as stated:

Line 483-484 (438): *"All basal melt maps are available at the GEUS Dataverse website (<https://dataverse01.geus.dk/>). DOI: 10.22008/FK2/PLNUJO."*

Reviewer #2 (Remarks to the Author):

Karlsson et al.,

Overall, the authors have done a great job addressing some of my earlier concerns regarding the uncertainty in GHF. The inclusion of a full-Stokes model and 13 regional climate models are great improvements and frankly, it is the best we can do with our current modeling and observational capabilities. I only have stylistic suggestions and some nitpicking, minor comments to add.

Line comments:

L27: The main difference between altimetry and gravimetry is that GRACE does not require any assumptions about firn densification etc. in the height-to-area-to-volume conversion; GRACE directly measure mass. So I would suggest rewording this sentence to highlight that more, something like "directly" measuring "mass" changes "using" "gravimetry".

The sentence has been changed to: line 27-28: *"[...] by directly measuring mass changes using gravimetry[3]"*

L30-31: Stylistic suggestion: the average mass balance of the ice sheet "between 2005-2015" is ... And then you can get rid of the use of parentheses in the end.

Changed as suggested.

L35: Nitpicking but you can be more specific here and say: accurate representation of the "climatic and dynamic" mass loss terms.

Changed as suggested.

L49: Add a comma before "while"

Changed as suggested.

L51: I'd specify that "surface melt input to the bed" as opposed to surface melt production that matters to heat transfer to the basal melt component.

Changed to line 50-51 “[...] and heat from surface melt input to the bed, vary in response to changes in ice dynamics and surface melt, respectively.”

L52: Remove “the” after quantify.

Changed as suggested.

L53: Remove “as well as”

Changed as suggested.

L57: Add “or land-margin” since you considered both marine and land-terminating glaciers here.

Changed as suggested.

L58: I'm not sure there is an absolutely 0% chance that large subglacial lakes exist in Greenland (e.g. Oswald & Gogineni, 2012 seems to think there are plenty in the ice sheet interior), but I think it is likely that there is a limited "long-term" storage of meltwater in subglacial lakes given the high input of surface meltwater fluxes and geometric effects of the ice sheet that you have highlighted. So may I suggest we reword this sentence something along the lines: preclude the existence of “long-term meltwater storage in” large subglacial lakes?

Changed to: line 57-59 “We assume that all basal melt water is discharged to the ocean or land-margin since the geometry and high surface slopes of the ice sheet preclude the existence of long-term meltwater storage in subglacial lakes[19]”.

L78: Is it “evenly distributed”? Looking at Fig. 1D, I'd say geothermal basal melt is quite spatially variable between the ice sheet interior and lower elevation regions.

Reviewer 1 also picked up on this inconsistency. The sentence has been changed to line 79: “unevenly distributed”.

L85: Remove “The” before frictional heat. And same for the next sentence before frictional heat.

Changed as suggested.

L92: Stylistic suggestion: Note that Elmer/Ice predicts...

Changed as suggested.

L99: Remove “the” before geothermal flux instead say “geothermal heat fluxES”.

Changed as suggested.

L103: Stylistic suggestion: replace “, and” with “with”

The sentence has been changed to line 107: “[...] with rates exceeding 0.2m per year close to the margin.”

L107: Add more information here by saying “.... important in slow-moving sectors in areas that on average move at XXX myr⁻¹ or less”.

The sentence following this has been modified to provide information and context the flow of the SW sector.

Line 111-113 (107): *In the predominantly land-terminating southwestern (SW) sector, where average velocity is 45 m per year compared to the 61 m per year Greenland-wide average, friction melt does not exceed 0.2 m per year except in a few grid cells by the ice margin.*

L111: largest “and fastest”

Changed as suggested.

L115: Remove “the” before “heat”

Changed as suggested.

L119: Instead of using a single 2000 m cutoff for all of Greenland, why didn't you use the ELA to capture the impact of spatial difference in the ablation zone area? (assuming we only have top-to-bottom water transport in the ablation zone)

We use the 2000m cutoff because that is the elevation where meltwater is likely to reach bedrock. In areas with thicker ice, crevasses or moulins, water is less likely to penetrate to the bed and thus the surface meltwater cannot enter the subglacial system. The ELA location is indirectly included via the RCM. In Mankoff and Tulaczyk (2017), the authors explored the impact of using ELA instead of 2000m on basal melt and found that the resulting change was less than 5%.

We have added the following sentences:

Line 472-475: [...] *en- and subglacial conduits*[54]. *Note that we also assume that the water only penetrates to the bed at altitudes below 2000 m above sea level. Tests using equilibrium line altitude instead of the 2000 m elevation contour found that the resulting change in basal melt was less than 5 %*[34] .

L131: Remove “the” before melt

Changed as suggested.

L132: Remove “the” before conduits

Changed as suggested.

L172: Mention it's an underestimation because you are not using summer surface velocity.

We have added the following:

Line 181-182 (172): *“The use of winter velocities entails that we are underestimating the friction heat while the simplified approach introduces additional uncertainties (see methods).”*

L175: Any thoughts as to why the Full-Stokes model gives a lower estimate than the simplistic approach, especially in NE and NW sectors? Is it because Elmer/Ice is missing some complicated feedback mechanisms or something else?

It is most likely an overestimate from the shallow-ice approximation rather than Elmer/Ice underestimating. The results are consistent with Maier et al. (2020) who shows that, even when averaged over 6km grid cells, the simple approach overestimates the basal stresses compared to the Full-Stokes solution. The difference is particularly important in the western sectors that have numerous small outlet glaciers with complex flow and topography. In NE sector, the shallow-ice approximation cannot capture the

complex flow of the Northeast Greenland ice stream. We have added the following sentence to the manuscript:

Line 185-191 (175): *"[...] with the largest differences in the NE region (59%) and NW sector (52%) (see methods and supplementary materials). The reason for the large discrepancies is likely the inability of the simplified approach to capture the complex flow regime of the Northeast Greenland ice stream in the NE sector, and the topography of numerous small outlet glaciers in the NW sector. Our findings are consistent with a recent study showing that the simple approach overestimates the basal stresses compared to the Full-Stokes solution[32]."*

L179: remove "the" after "include" and "the" before "basal shear layer" and turn "uncertainty" into plural since there are multiple sources of uncertainty in velocity.

We have removed "the" after "include" but not before "basal shear layer". Uncertainty has been changed to plural as suggested.

L220: Again, I believe there is a distinction between increased surface melt production vs melt input to the bed here. While melt production will likely increase under warming climate scenarios with both increasing melting in the ablation zone and migration of surface ablation to higher elevations, we may also see an increase in surface runoff, e.g. if meltwater refreezes in firn forming a low permeability barrier that reduces vertical percolation (e.g. MacFerrin et al., 2019). So I'd caution putting in a broad-stroke statement like this, and maybe reword this to something more like "heat transported by surface meltwater will increase with greater meltwater production, which will likely increase meltwater delivery to the bed especially in the ablation zone."

Changed as suggested.

REVIEWERS' COMMENTS

Reviewer #1 (Remarks to the Author):

I appreciate the detailed response the authors made to my last round of comments. In particular, the description of the methods for the viscous heat dissipation term has been greatly improved. I have no additional concerns about the manuscript and think it will be a valuable contribution to current understanding of the mass balance of the Greenland Ice Sheet.

Dear editor Dr. Kasey Bolles,

Thank you for guiding the review process of our manuscript. We thank the reviewers the detailed and thorough comments they have provided throughout. We note that the comments from the reviewer (inserted below) do not require further changes.

We have made the following two minor changes:

- The section on data availability (lines 483-485) now reads: "*All basal melt maps are available at the GEUS Dataverse, DOI: 10.22008/FK2/PLNUEO [55].*" Where [55] refers to the data citation included in the references.
- The section on code availability (lines 488-491) now reads: "*Code showing examples of how to generate Figures 1D, 1E, 1F and 2A are available at the GEUS Dataverse website [55].*" This change was implemented because we are in the process of moving repositories on Github.
- Figure captions have been changed to include a brief title that summarises the whole figure and the labels have been changed to lower case.
- Figure 3: Caption has been changed to: "*...from winter 2000/2001 through to summer 2019.*"
- References to supplementary material is changed to Supplementary Note X or Supplementary Figure Y, as appropriate.

All changed in the main text has been marked with red.

Our sincere thanks to both reviewers for their constructive and helpful feedback and comments.

Best,

Nanna B. Karlsson on behalf of all co-authors.

REVIEWERS' COMMENTS

Reviewer #1 (Remarks to the Author):

I appreciate the detailed response the authors made to my last round of comments. In particular, the description of the methods for the viscous heat dissipation term has been greatly improved. I have no additional concerns about the manuscript and think it will be a valuable contribution to current understanding of the mass balance of the Greenland Ice Sheet.

Reply: Thank you.